# GPROF V7 and beyond: Assessment of current and potential future versions of the GPROF passive microwave precipitation retrievals against ground radar measurements over the continental US and the Pacific Ocean

Simon Pfreundschuh[1, 3], Clément Guilloteau[2], Paula J. Brown[1], Christian D. Kummerow[1], and Patrick Eriksson[3]

[1]Department of Atmospheric Science, Colorado State University, Fort Collins, CO 80523, United States of America
[2]Department of Civil and Environmental Engineering, University of California, Irvine, Irvine, California
[3]Department of Space, Earth and Environment, Chalmers University of Technology, 41296 Gothenburg, Sweden

**Correspondence:** Simon Pfreundschuh (simon.pfreundschuh@colostate.edu)

**Abstract.**

The Goddard Profiling Algorithm (GPROF) is used operationally for the retrieval of surface precipitation and hydrometeor profiles from the passive microwave (PMW) observations of the Global Precipitation Measurement (GPM) mission. Recent updates have led to GPROF V7, which has entered operational use in May 2022. In parallel, development is underway to
improve the retrieval by transitioning to a neural-network-based algorithm called GPROF-NN.

This study validates retrievals of liquid precipitation over snow-free and non-mountainous surfaces from GPROF V7 and multiple configurations of GPROF-NN against ground-based radar measurements over the conterminous United States (CONUS) and the tropical Pacific. GPROF retrievals from the GPM Microwave Imager (GMI) are validated over several years and their ability to reproduce regional precipitation characteristics and effective resolution is assessed. Moreover, the retrieval accuracy
for several other sensors of the constellation is evaluated.

The validation of GPROF V7 indicates that the retrieval produces reliable estimates of liquid precipitation over CONUS. During all four assessed years, annual mean precipitation is within $8\%$ of gauge-corrected radar measurements. Although biases of up to $25\%$ are observed over sub-regions of the CONUS and the tropical Pacific, the retrieval reliably reproduces each region's diurnal and seasonal precipitation characteristics. The effective resolution of GPROF V7 is found to be $51\,\mathrm{km}$
over CONUS and $18\,\mathrm{km}$ over the tropical Pacific. GPROF V7 produces robust precipitation estimates also for the other sensors of the GPM constellation.

The evaluation further shows that the GPROF-NN retrievals have the potential to significantly improve the GPM PMW precipitation retrievals. GPROF-NN 1D, the most basic neural network implementation of GPROF, improves the mean-squared error, mean absolute error, correlation and symmetric mean absolute percentage error of instantaneous precipitation estimates
by about twenty percent for GPROF GMI while the effective resolution is improved to $31\,\mathrm{km}$ over land and $15\,\mathrm{km}$ over oceans. The two GPROF-NN retrievals that are based on convolutional neural networks can further improve the accuracy up to the level of the combined radar/radiometer retrievals from the GPM core observatory. However, these retrievals are found to overfit

on the viewing geometry at the center of the swath, reducing their overall accuracy to that of GPROF-NN 1D. For the other sensors of the constellation, the GPROF-NN retrievals produce larger biases than GPROF V7 and only GPROF-NN 3D achieves consistent improvements compared to GPROF V7 in terms of the other assessed error metrics. This points to shortcomings in the hydrometeor profiles or radiative transfer simulations used to generate the training data for the other sensors of the GPM constellation as a critical limitation for improving GPM PMW retrievals.

## 1 Introduction

Satellite-derived, global measurements of precipitation are essential for a wide range of societal and scientific applications, including the identification of climatic changes in the hydrological cycle (Hegerl et al., 2015), the validation of climate models (Tapiador et al., 2017) as well as the management of droughts (Kirschbaum et al., 2017) and risks associated with extreme weather (Skofronick-Jackson et al., 2017). The Global Precipitation Measurement (GPM, Hou et al., 2014) aims to provide such measurements at high spatial and temporal resolution. The GPM core observatory, the core satellite of the mission, carries the Dual-frequency Precipitation Radar (DPR) and the GPM Microwave Imager (GMI). The combined observations from DPR and GMI are used to produce global reference retrievals of precipitation and hydrometeor profiles (Grecu et al., 2016). These reference measurements are used to produce the a priori or training data for precipitation retrievals from a constellation of passive microwave (PMW) sensors. Finally, retrieval results from the combined DPR/GMI and PMW observations are combined with data from geostationary satellites and rain gauges to produce the level 3 GPM IMERG product (Huffman et al., 2020), which provides global precipitation estimates at a nominal spatial resolution of $0.1°$ and a temporal resolution of $30\,\mathrm{min}$.

The Goddard Profiling Algorithm (GPROF, Kummerow et al., 2015) is the algorithm used for the operational processing of the PMW precipitation retrievals from GPM. Since the PMW retrievals account for the bulk of the precipitation retrievals integrated into GPM IMERG, GPROF plays a central role in the GPM processing pipeline. Recent algorithm development has produced a new version of the retrieval, GPROF V7, which replaced GPROF V5 in the operational processing of the GPM precipitation products in May 2022. In parallel, two novel, neural-network-based implementations of GPROF have been developed. These implementations, GPROF-NN 1D and GPROF-NN 3D, have been assessed under idealized conditions in Pfreundschuh et al. (2022) where it was found that they can significantly improve the accuracy of the GPM PMW precipitation retrievals. An analysis of the spatial variability in the retrieved precipitation fields shows that the GPROF-NN retrievals notably improve the effective resolution of the retrieval, i.e., increase its ability to reproduce the small-scale spatial variability of precipitation. Given the considerable improvements afforded by the neural-network-based implementations of GPROF, integrating these algorithms into the operational processing presents a simple and cost-effective opportunity to improve global precipitation measurements.

The GPROF and GPROF-NN algorithms have in common that they are designed to reproduce precipitation estimates from a dataset of reference precipitation measurements, the so-called *a priori database*. Climatologically, precipitation estimates from GPROF and GPROF-NN will be similar to those of the a priori database. If the a priori database deviates from given validation measurements, the resulting errors will be reproduced by the GPROF and GPROF-NN retrievals. Pfreundschuh et al. (2022)

evaluated the GPROF retrievals against held-out data from the a priori database. In this scenario, all retrieval errors are due to the limited information content of the input observations and the limitations of the retrieval method.

This study compares GPROF retrievals to independent validation data derived from ground-based precipitation radars. In this case, differences between a priori database and the validation data constitute a second source of errors that will increase the total retrieval error. These two sources of error are fundamentally different and reducing their impact requires different approaches. Therefore, quantifying the extent to which these sources contribute to the total retrieval error is essential to guide future efforts to improve GPM PMW retrievals.

This study validates the newly-released operational version of GPROF and the GPROF-NN algorithms against independent, ground-based precipitation measurements. The questions that this study aims to address are:

1. What are the changes and improvements in GPROF V7 compared to GPROF V5?

2. What is the contribution of errors in the a priori database to the GPROF retrieval error?

3. Can the GPROF-NN retrievals improve precipitation estimates of the GPM PMW observations when compared to independent measurements?

The validation is based on several years of collocations between different GPM sensors and ground-based precipitation radars over the conterminous United States (CONUS) and the Kwajalein atoll in the tropical Pacific. Our analysis focuses on liquid precipitation estimates over snow-free and non-mountainous surfaces. The principal motivation this is that PMW precipitation estimates of frozen precipitation and over snow-covered and mountain surfaces are particularly uncertain, and, since they constitute only a minor part of the total validation data, corresponding larger retrieval errors may not be reflected in the overall validation statistics.

The first part of this study focuses on retrievals from GMI, for which a detailed evaluation of the retrieval accuracy from instantaneous to annual time scales is performed. In addition, we analyze the effective resolution of the GMI retrievals to confirm the ability of the GPROF-NN retrievals to improve the effective resolution of GPROF, which has been found to be particularly low over land (Guilloteau et al., 2017). The second part of the study assesses the retrieval accuracy of a selection of currently operational and historical sensors of the GPM constellation. Finally, we assess the impact of limiting this validation to liquid precipitation and retrievals over snow-free and non-mountainous areas.

## 2  Data and methods

This investigation uses quantitative precipitation estimates (QPEs) from ground-based radars as principal means for validating the GPROF retrievals. While these measurements are themselves affected by non-negligible uncertainties, they offer the best compromise in terms of sensitivity to precipitation and ability to resolve the spatial structure of precipitation fields. Gauge-corrected radar estimates will be used where possible as they have been shown to reduce uncertainties (Kirstetter et al., 2012). Since the ground-based radar QPEs are used as independent measurements to validate the GPROF retrievals, we will refer to them as *validation measurements or validation data*.

Due to the importance of the a-priori database for the GPROF retrievals, precipitation estimates from the a priori database and the GPM combined radar/PMW retrievals (upon which large parts of the a priori database are based) are included in the assessment of the GPROF retrievals. To distinguish these measurements from the validation measurements and since the GPROF retrievals aim to reproduce these estimates, they will be referred as *reference data*. Comparing the accuracy of GPROF with respect to validation and reference data as well as the accuracy of the reference data with respect to the validation data will allow attributing retrieval errors to the observation system and retrieval method on one hand and the a priori/training database on the other.

## 2.1 Validation data

### 2.1.1 MRMS

The principal source of validation measurements for this study are instantaneous, gauge corrected precipitation estimates from the NOAA Multi-Radar Multi-Sensor System (MRMS). These estimates are produced specifically for GPM ground validation and are gauge-corrected to match monthly accumulations following the approach described (Kirstetter et al., 2012). These estimates are not part of the operational MRMS processing suite but can be obtained from the GPM ground validation data archive (Wolff, 2023). The processing of the ground-validation data includes a basic filtering that removes measurements with excessive gauge-correction factors (Kirstetter et al., 2012). The data is provided on an approximately $0.01\,° \times 0.01\,°$ grid covering the CONUS. For the comparison against the satellite retrievals, the MRMS data is smoothed using a Gaussian average filter with a full-width at half-maximum (FWHM) of $5\,km$. Following this, the mapping to the collocation grid is performed using nearest-neighbor interpolation.

The MRMS validation data includes a radar quality index (RQI), which quantifies the reliability of the MRMS measurements. It takes into account beam blocking, beam height and width, and the height of the melting layer. The RQI is mapped to the collocation grid using nearest neighbor interpolation. The precipitation mask is downsampled by applying a Gaussian averaging filter with a FWHM of $5\,km$ to the binary mask of each class. The resulting fields are interpolated to the $5\,km \times 5\,km$ collocation grid and each grid point is assigned the class with the highest mask value.

### 2.1.2 Kwajalein Polarimetric S-band Weather Radar

Validation data for the Pacific ocean stems from the Kwajalein Polarimetric S-band Weather Radar (KPOL). The lowest-elevation scan is used to estimate surface precipitation. A full $360\,°$ sweep consists of 360 scans, each of which comprises 779 radar bins with a range resolution of $200\,m$. Rain rates are estimated using the CSU-HIDRO algorithm for polarimetric radar observations (Cifelli et al., 2011). The radar has recently been calibrated using the self-consistency method proposed by Ryzhkov et al. (2005).

## 2.2 Reference data

### 2.2.1 GPM combined

Surface precipitation from version 7 of the GPM combined radar/radiometer product (GPM CMB, Grecu et al., 2016) accounts
for the major part of the surface precipitation in the GPROF a priori database over ocean and snow-free land surfaces. It is
included in this validation as a proxy for the retrieval database in the years that are not covered by the a priori database.

### 2.2.2 A priori database

The GPROF V7 and GPROF-NN retrievals rely on a common *a priori database*, which comprises PMW observations and
corresponding hydrometeor profiles and precipitation rates. Over oceans and snow-free land surfaces, the database is derived
from collocations of GMI observations with corresponding surface precipitation and hydrometeor profiles from GPM CMB
(Grecu et al., 2016). Although GPM CMB also incorporates PMW observations from GMI, the retrieval is primarily driven by
measurements from the precipitation radar.

    Over oceans, light precipitation from the currently operational Microwave Integrated Retrieval System (MIRS, Boukabara
et al., 2011) is added to pixels in which no precipitation is detected by the GPM CMB product. Over mountain regions, the
precipitation from the combined product is increased to account for orographic effects. Over snow surfaces, the database uses
collocations between GMI and MRMS, while precipitation over sea-ice is derived using collocations with the ERA5 reanalysis
(Hersbach et al., 2020). The current GPROF retrieval database used by GPROF V7 and the GPROF-NN retrievals is based on
observations from the water year 2019.

    The GPROF V5 and V7 retrievals cluster the a priori database based on the similarity of the observations in order to reduce
the computational complexity of performing the retrieval. However, both the training of the GPROF-NN retrievals as well as
the analysis presented here use the unclustered database.

## 2.3 GPROF retrievals

### 2.3.1 GPROF V7

As of May 2022, the publicly available version of GPROF is V7. GPROF V7 adds several algorithm improvements over GPROF
V5, the previous operational version. Most notably, new surface classes have been introduced to account for orographic effects.
GPROF V5 is included in the validation as baseline retrieval against which GPROF V7 and the GPROF-NN algorithms will be
assessed. Since GPROF V7 replaced GPROF V5 in the spring of 2022 as the operational version of the GPM PMW retrievals,
comparisons to GPROF V5 are only possible for the water years 2019, 2020 and 2021.

### 2.3.2 GPROF-NN

GPROF-NN (Pfreundschuh et al., 2022) is a novel, neural-network-based implementation of GPROF. The GPROF-NN retrieval
has been designed as a drop-in replacement for GPROF V7 and is based on the same a priori database as GPROF V7. Two

principal configurations of GPROF-NN exist, named GPROF-NN 1D and GPROF-NN 3D, which differ in their treatment of the input observations. While GPROF-NN 1D retrieves precipitation independently for each observed pixel, GPROF-NN 3D uses all pixels in the swath simultaneously, thus enabling the retrieval to leverage structural information in the PMW observations. The implementation of both retrievals are available through a public code repository (Pfreundschuh, 2022).

### 2.3.3 GPROF-NN HR

All retrieval variables in the GPROF database are spatially smoothed to the expected resolution of the respective sensor. For GMI, the expected resolution is $18.7\,\mathrm{km}$ in along track direction and $11\,\mathrm{km}$ in across track dimension, which corresponds to the FWHM of the $18.7\,\mathrm{GHz}$ channels. The higher frequency channels of GMI have smaller footprints and may thus allow for retrieving precipitation at higher resolution. However, the smoothing of the precipitation in the a priori database limits on the resolution of the retrieved precipitation to that of the $18.7\,\mathrm{GHz}$ channels.

To explore the possibility of achieving higher resolution retrievals from GMI, we include an additional configuration of the GPROF-NN retrieval, which is named GPROF-NN *high-resolution* (HR). GPROF-NN HR retrieves precipitation at a resolution of approximately $5\,\mathrm{km} \times 5\,\mathrm{km}$. It is trained directly on GPM CMB surface precipitation without any smoothing applied to the precipitation fields. Furthermore, it does not make use of the ancillary data used by GPROF V7 and the other two GPROF-NN versions described above. This has the advantage that the retrieval can be run directly on the level 1C-R observations from GMI. The underlying neural network model uses the same architecture as the GPROF-NN 3D retrieval adapted to the reduced number of inputs. The output resolution of $5\,\mathrm{km} \times 5\,\mathrm{km}$ is achieved through an additional upsampling block in the final part of the decoder, which increases the resolution in the along-track direction by a factor of 3. The source code of the GPROF-NN HR retrieval is published together with the other GPROF-NN retrievals (Pfreundschuh et al., 2022).

## 2.4 Collocations

For the comparison of validation and reference measurements and the GPROF retrievals, all data have to be collocated. To this end, ground- and satellite-based results are remapped to a common, approximately equidistant $5\,\mathrm{km} \times 5\,\mathrm{km}$ grid that follows the satellite swath. An equidistant grid is required for the analysis of the effective resolution. The grid resolution of $5\,\mathrm{km} \times 5\,\mathrm{km}$ was chosen finer than the expected resolution of the PMW-based GPROF retrievals in order to avoid interference of the spatial sampling with the analysis of the effective resolution of the retrievals.

An example of the collocations used in this validation study is provided in Fig. 1. The scene displays an overpass of the GPM core observatory over the mid-latitude cyclone associated with the 2020 Easter tornado outbreak (Kerr and Alsheimer, 2022). The various panels show surface precipitation estimates from the validation and reference measurements, and the different GPROF retrieval results.

Four water years (October through September) from 2019 until 2022 are used for the validation of the GPROF retrievals from GMI. This time period was chosen because it includes the water year 2019, which is used to derive the a priori database for the GPROF retrievals. The analysis is extended to include several years in order to allow assessment of the temporal variability of the retrieval accuracy.

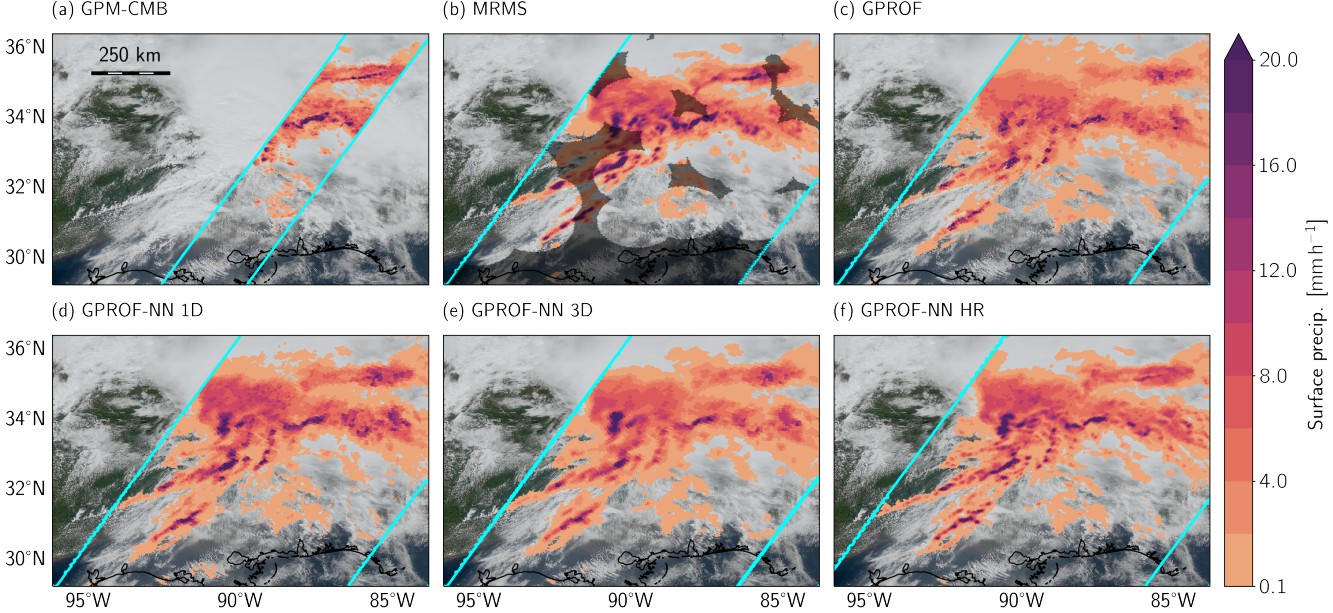

**Figure 1.** Sample scene from the collocation dataset used to validate the GPROF retrievals displaying an overpass of the GPM core observatory over the 2020 Easter Tornado outbreak. Results are shown for the GPM CMB product (Panel (a)), gauge-corrected MRMS QPEs from ground-based radars (Panel (b)), the GPROF V7 retrieval (Panel (c)), the GPROF-NN 1D retrieval (Panel (d)), the GPROF-NN 3D retrieval (Panel (e)) and the GPROF-NN HR retrieval (Panel (f)). Cyan lines in Panel (a) mark the boundaries of the DPR, while the cyan-colored lines in the remaining panels mark the boundaries of the GMI swath. Dark shading in Panel (b) marks measurements with a RQI lower than 0.8.

For the other sensors of the GPM constellation, the retrieval accuracy was assessed using a single year of collocations. The collocation periods were chosen according to the availability of observations and ground-validation data. The periods chosen for the different sensors are listed in Table 1.

| Sensor | Platform | Collocation period |
|--------|----------|--------------------|
| TMI    | TRMM     | 2013-03-01 — 2014-03-01 |
| SSMIS  | F17      | 2015-01-01 — 2016-01-01 |
| AMSR2  | GCOM-W1  | 2018-10-01 — 2019-10-01 |
| MHS    | NOAA-19  | 2018-10-01 — 2019-10-01 |

**Table 1.** Other GPM sensors and corresponding collocation periods used to evaluate the GPROF retrievals.

## 2.5 Analysis of the effective resolution

The distance between the centers of neighboring pixels of the GMI swath is approximately $13.5\,\mathrm{km}$ in along-track direction and $5\,\mathrm{km}$ in across-track directions at the center of the swath. Since GPROF traditionally retrieves precipitation at the same grid as the input observations, the resolution of the retrieved precipitation fields cannot exceed the spatial sampling of the input observations. In general, however, not all variability at scales larger than the spatial sampling of the output data is truthfully represented in the retrieval results. This motivates the definition of an effective resolution as a numerical measure of retrieval's ability to resolve small-scale spatial variability.

The consistency between retrieved and reference precipitation fields across different spatial scales can be quantified using the relative spatial Fourier spectral coherence between reference and retrieved fields. The spectral coherence can be interpreted as the complex correlation between the Fourier coefficients of two signals within a narrow frequency (or wavenumber) band. In two dimensions, the Fourier spectral coherence between two fields $g(x,y)$ and $h(x,y)$ is defined as:

$$C_{g,h}(k_x,k_y) = \frac{P_{g,h}(k_x,k_y)}{\sqrt{P_g(k_x,k_y) \times P_h(k_x,k_y)}} \tag{1}$$

with $k_x$ and $k_y$ the wavenumbers in the two orthogonal directions $x$ and $y$. $P_g(k_x,k_y)$ is the Fourier power spectral density (PSD) of $g(x,y)$, defined as:

$$P_g(k_x,k_y) = \hat{g}(k_x,k_y) \times \hat{g}^*(k_x,k_y) \tag{2}$$

with $\hat{g}^*$ denoting the complex conjugate of the Fourier coefficient field $\hat{g}$. $P_{g,h}(k_x,k_y)$ is the Fourier cross power spectral density (CPSD) between $g(x,y)$ and $h(x,y)$ defined as:

$$P_{g,h}(k_x,k_y) = \hat{g}(k_x,k_y)\hat{h}^*(k_x,k_y) \tag{3}$$

The computation of the PSD and CPSD is performed through a discrete cosine transform (DCT) and averaged over the evaluated retrieval scenes.

Passing into the polar coordinate system through the change of variables $k = \sqrt{(k_x^2 + k_y^2)}$ and $\theta = \mathrm{atan}(\frac{k_x}{k_y})$, the spectral quantities can be analyzed as a function of the spatial wavenumber $k$ and the azimuthal direction $\theta$. Since we did not find any significant dependency on the azimuthal direction, we integrated the spectral quantities along $\theta$ to analyze the coherence as a function of $k$ only.

The effective resolution of the retrieved fields can then be defined from the spectral coherence between retrieved and reference field as

$$\mathrm{ER} = \frac{1}{2}\arg\max_k C_{g,h}(k) \geq \frac{1}{\sqrt{2}}, \tag{4}$$

i.e. half of the shortest wavelength for which the spectral coherence is larger than $\frac{1}{\sqrt{2}}$. Defined in this way, the effective resolution corresponds to the smallest spatial scale at which the signal to noise ratio of the retrieved precipitation fields exceeds 1.

## 3 Results

The assessment of the GPROF retrieval against ground-based radar is split into four parts. The first part analyzes the accuracy of the GMI-based precipitation estimates over CONUS and the Kwajalein site, while the second part assesses the effective spatial resolution of these retrievals. The third part evaluates the retrieval accuracy for other sensors of the GPM constellation of PMW radiometers for which MRMS ground-validation data is available. Finally, the fourth part analyzes the contributions and retrieval biases of frozen precipitation and over snow-covered and mountain surfaces, which are excluded from most analyses in the previous sections.

### 3.1 Retrieval accuracy of GPROF for GMI

The validation of the GPROF GMI retrievals uses collocations from October 2018 until October 2022. The first year of this period coincides with the a priori database. For the comparison against MRMS, only data points with an RQI of at least 0.8 were used. The validation is performed at the $5\,\mathrm{km}$ resolution of the collocations. We choose to evaluate the retrieval at this resolution instead of the footprint size of the PMW observations because it makes accuracy metrics between different sensors easier to compare.

Retrievals over snow-covered and mountain surfaces are excluded from the validation due to the uncertainties in both the satellite estimates as well as the validation data. In addition to this, the GPROF a priori database for snow-covered surfaces is derived from collocations with MRMS, while precipitation over mountains is obtained by scaling the GPM CMB precipitation to account for the orographic enhancement of precipitation. These two modifications aim to counteract known weaknesses of the GPM CMB retrievals, but would skew the comparison between GPM, GPM CMB, and MRMS.

Similarly, precipitation that is identified as frozen by MRMS is excluded from the validation. The retrieval of frozen precipitation from both PMW and radar is particularly challenging due to its uncertain radiometric properties. Because of these increased uncertainties and the small contribution of frozen precipitation to the total precipitation in the validation data, the retrieval accuracy for frozen precipitation should be assessed in a separate, dedicated study.

### 3.1.1 CONUS

The conditional distributions of retrieved instantaneous precipitation conditioned on the corresponding precipitation from the a priori/training database and validation measurements from MRMS are displayed in Fig 2. The first column of panels shows the distribution of retrieved precipitation with respect to the GPROF a priori/training database. The spread in these distributions is due to the limitations of the retrieval method and the ill-posed character of the retrieval and they thus represent the best-case accuracy of the GPROF and GPROF-NN retrievals. Some spread is observed even between GPM CMB and the a priori database. This is due to the spatial smoothing applied to the precipitation in the a priori database, which causes precipitation measurements in the a priori/training database to have lower resolution than GPM CMB. The difference in the resolution between the GPROF(-NN) retrievals and GPM CMB with respect to the a priori database also explains the opposing behavior of the respective conditional means at light and heavy precipitation rates.

The spread in all GPROF retrievals is significantly larger when compared to MRMS. However, even the GPM CMB retrieval shows significant deviations from the MRMS measurements. From the GPROF retrievals, V5 and V7 exhibit the largest spread, while there is less spread in the distributions of the GPROF-NN 1D and 3D retrievals. Compared to the 1D and 3D variants, the GPROF-NN HR retrieval further decreases the spread. The distributions corresponding to different validation periods are very similar for all retrievals. An exception to this is a notable increase in the overestimation of light precipitation after the water year 2020 that affects all retrievals. As will be discussed below, this coincides with a change in the regional biases and is likely due to a change in the processing of the MRMS esimates (Anonymous Referee 2, 2023). Apart from this, however, the accuracy of each retrieval conditioned on the MRMS validation precipitation exhibits little inter-annual variability.

Maps of annual mean retrieval biases, calculated as the annual average of the difference between retrieved precipitation and the precipitation in the a priori database or the MRMS validation data, are displayed in Fig. 3. When evaluated against the a priori/training database, GPROF V5 exhibits the largest biases. Since GPROF V5 is based on a different a priori database, which, in contrast to the a priori database of GPROF V7, uses estimates from GPM DPR-Ku over land, this is expected. However, even GPROF V7 exhibits noticeable dry biases along the east and west coasts and a weaker wet bias over the central CONUS. GPROF-NN 1D exhibits a weaker dry bias over the western CONUS and the South-East and a wet bias in the North-East. The biases of the GPROF-NN 3D and GPROF-NN HR retrievals are even weaker but exhibit a spatially more consistent tendency to overestimate precipitation. Since the a priori/training database is derived from GPM CMB, GPM CMB is practically bias-free compared to the retrieval database.

Significantly larger biases are observed when the precipitation is compared to MRMS measurements. For the year 2019, the GPM CMB retrieval exhibits a wet bias over the western Great Plains and a weak dry bias over the remaining CONUS. A similar pattern is observed for all the GPROF retrievals, indicating that the bias patterns of the GPROF(-NN) retrievals result from biases in the a priori/training database.

Dry biases in the South-East and North-East persist throughout all considered years. Over the western CONUS, however, the retrievals are biased dry during years 2019 and 2020 but biased wet in the years 2021 and 2022. As pointed out by one of the anonymous reviewers, it is likely that this is due a change in the gauge correction methodology that occurred around October 2020 (Anonymous Referee 2, 2023). The spatial distributions of retrieval biases for the GPROF V7 and GPROF-NN retrievals largely follow the biases of GPM CMB also for the years 2020, 2021 and 2022. Given that the CMB over land is largely a radar-derived product, it is possible that the bias relative to MRMS is introduced by the rain gauge correction applied to the validation data and caused by precipitation properties that may not be resolved by the radar observations.

At the $0.5°$ resolution considered here, biases of GPROF V7 and the GPROF-NN retrievals are well correlated with those of GPM CMB and exhibit similar variability throughout the assessed years. A potential explanation for this are weather-regime-dependent errors in the GPM CMB retrieval with respect to the MRMS validation measurements. Since the GPM CMB retrievals are used to generate the a priori/training database of the GPROF and GPROF-NN retrievals, these retrievals reproduce the errors of GPM CMB, which leads to similar responses to the inter-annual variability of the occurrence of these weather regimes.

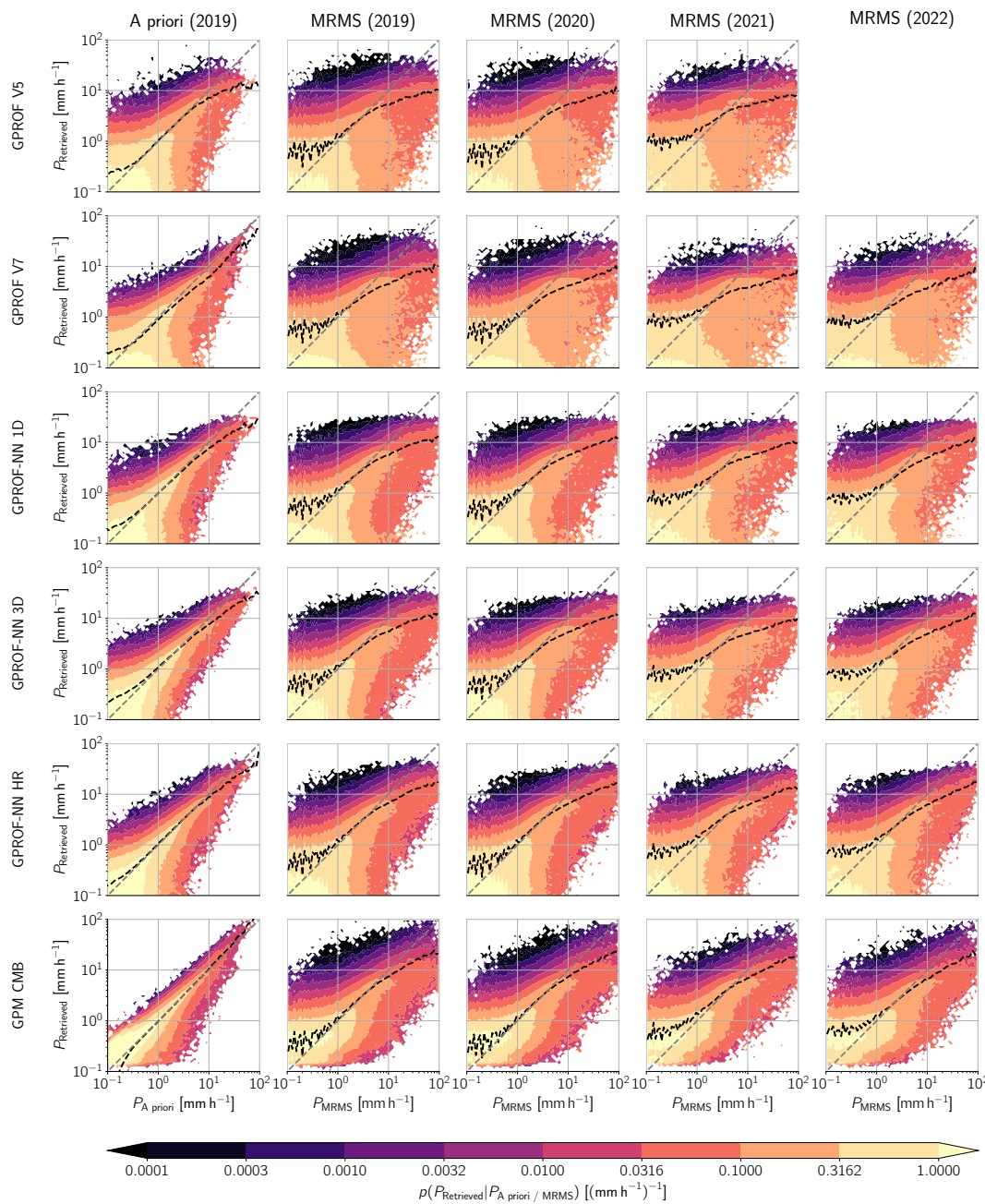

**Figure 2.** Distributions of retrieved precipitation ($P_{\text{Retrieved}}$) conditioned on precipitation from the a priori/training database ($P_{\text{A priori}}$) and MRMS validation data ($P_{\text{MRMS}}$). Panels in the left-most column show the distributions for all retrievals assessed against the GPROF V7 a priori database. The remaining columns show the respective distributions for the water years 2019, 2020, 2021, 2022 with respect to the MRMS validation QPEs. The black dashed line in each panel shows the conditional mean of the retrieved precipitation.

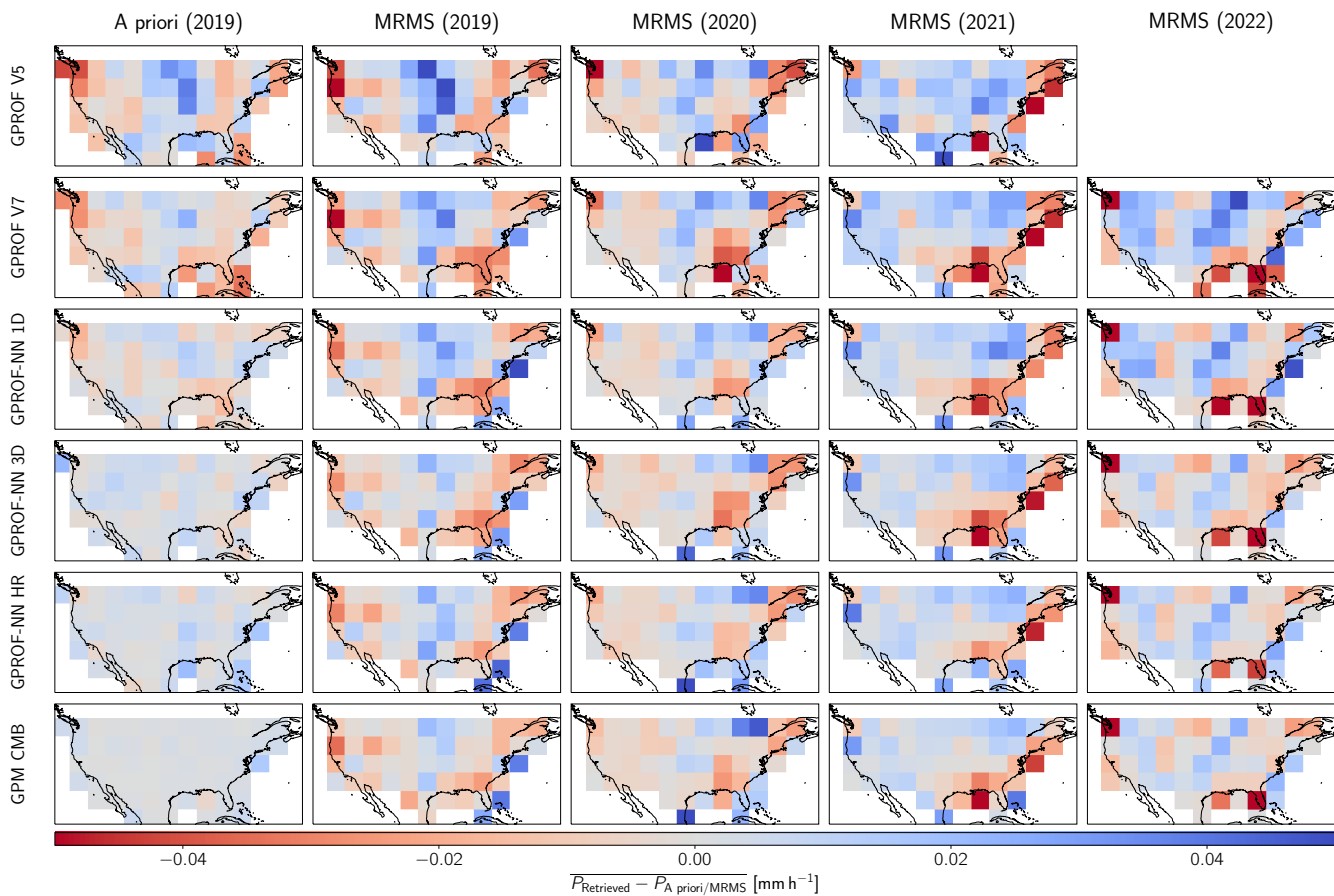

**Figure 3.** Spatial distribution of annual retrieval biases calculated as the sample mean of the difference between retrieved precipitation ($P_{\text{Retrieved}}$) and precipitation from the a priori/training database ($P_{\text{A priori}}$) and the MRMS validation data ($P_{\text{MRMS}}$). Each panel shows the biases calculated over a uniform longitude-latitude grid with a grid size of $5°$. Rows show the results for the different retrieval algorithms. The left-most column shows the biases with respect to the a priori/training database. Remaining columns show the biases with respect to the MRMS validation data for each of the assessed water years.

Maps of correlations between retrieved precipitation and precipitation from the a priori/training database and the MRMS validation data are displayed in Fig. 4. Since GPROF V5 is based on a different a priori database, it has the lowest correlation with respect to the a priori database. The correlation of GPROF V7 is highest in the western Central Plains and there even surpasses that of GPROF-NN 1D. However, GPROF-NN 1D achieves higher correlations than GPROF V7 in the remaining regions. Both GPROF-NN 3D and GPROF-NN HR achieve consistently higher correlations than GPROF-NN 1D. GPM CMB exhibits the highest correlation with the a priori database but remains below 1.0 due to the smoothing applied to the surface precipitation in the a priori database.

All retrievals have lower correlation with the MRMS QPEs than with the a priori/training database. In part, this reduction is due to the higher resolution of the MRMS validation precipitation compared to the reference precipitation in the a priori database. In addition to this, deviations of the a priori database from the MRMS validation data likely contribute to the reduced correlation. The reduction in correlation is strongest for the GPROF V7 retrieval, which exhibits only slightly higher correlations than V5 when evaluated against the ground-radar measurements. For the GPROF V5 and V7 retrievals correlations are between 0.4 and 0.6 over most of the CONUS. The correlations of GPROF-NN 1D are consistently higher, ranging from 0.5 to 0.7 in most regions. The correlations of GPROF-NN 3D are, again, slightly higher than those of GPROF-NN 1D and GPROF V7.

Surprisingly, the correlations of the GPROF-NN HR retrieval exceed those of GPM CMB. This is certainly remarkable, given that the GPROF-NN HR retrieval is trained on GPM CMB data. A possible explanation for this are random errors in the GPM CMB retrieval that are smoothed out in the passive-only retrievals.

A quantitative analysis of the retrieval accuracy is presented in Fig. 5. The assessed metrics include the mean error (Bias), mean-squared error (MSE), mean absolute error (MAE), correlation coefficient and the symmetric mean absolute percentage error (SMAPE). Several metrics are used to assess the retrieval accuracy in order ensure a comprehensive assessment of the quality of each algorithm's precipitation estimates. Definitions, basic characteristics and a motivation for the particular choice of metrics is provided in Appendix A.

All retrievals exhibit weak biases of the order of a few percent compared to the a priori/training database. When compared to MRMS, all retrievals except GPROF V5 exhibit a dry bias. The dry bias of GPM CMB increases from around $3\%$ for the water year 2019 to around $8\%$ in 2022. The GPROF-NN retrievals follow the tendency of the biases of GPM CMB with the 3D variant exhibiting a slightly stronger dry bias while the 1D and HR variants exhibit a slightly weaker dry bias. The biases of GPROF V7 follow the biases of GPM CMB for the years 2019, 2020, 2021 but decrease in 2022. GPROF V5 is closer to MRMS than the other retrievals for the years 2019, 2020, and 2021.

In terms of MAE, a consistent ranking of the retrievals can be observed. GPROF V5 has the largest MAE, closely followed by GPROF V7. GPROF-NN 1D, 3D, and HR yield consecutively lower MAEs and GPM CMB the lowest MAEs of all retrievals. Although the MAEs increase significantly when the retrievals are compared to MRMS measurements, the relative accuracy of the retrievals remains the same. Similar results are found for the MSE with the exception that the GPM CMB retrieval has higher MSE than the GPROF-NN HR retrieval.

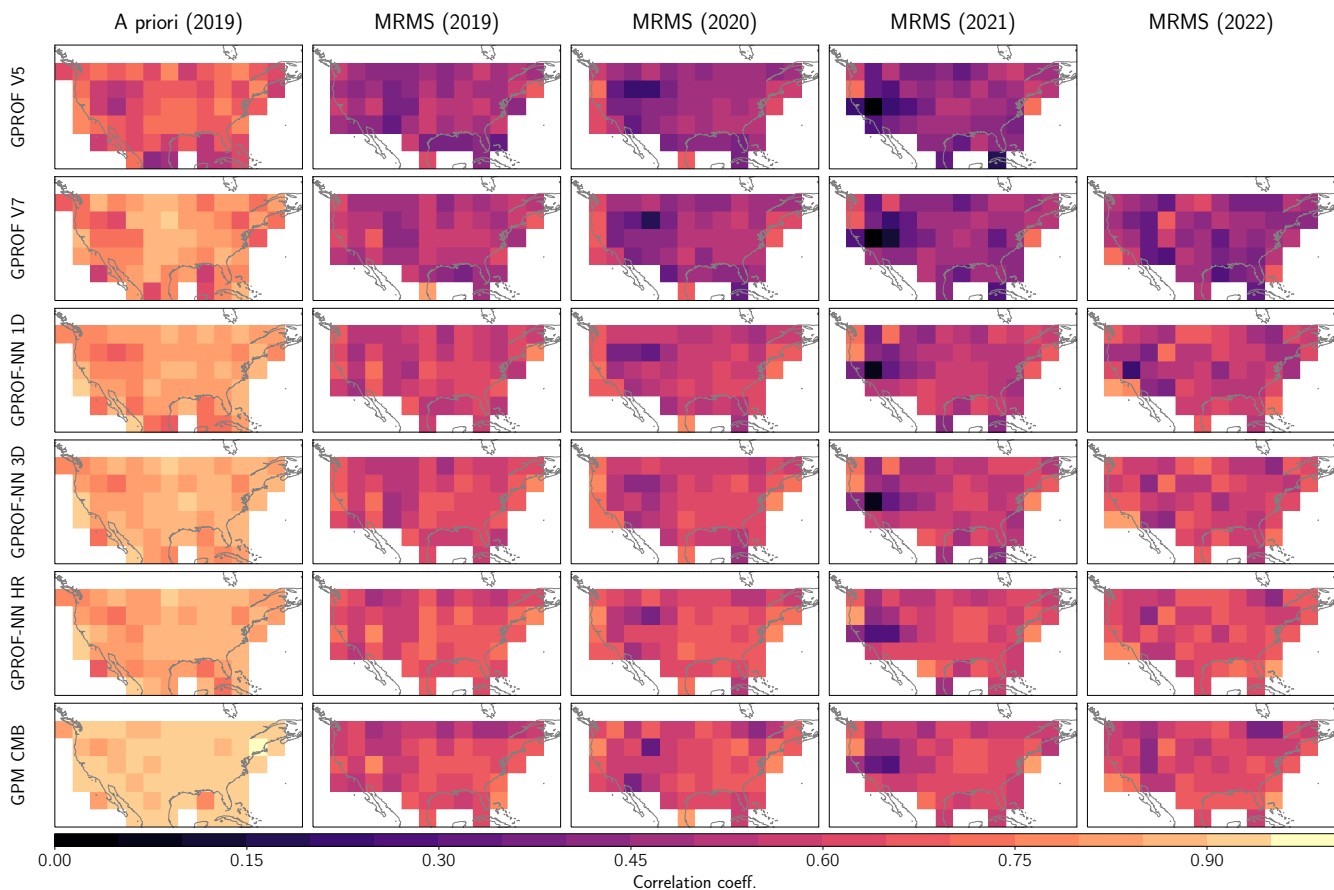

**Figure 4.** Like Fig. 3 but for the correlation coefficient between retrieved surface precipitation and a priori/training database and MRMS validation data.

The results for the correlation coefficient reflect the general findings from Fig. 4. With respect to the a priori database, GPROF V5 has the lowest correlation. The correlations of GPROF V7 and GPROF-NN 1D are almost identical and exceed those of GPROF V5 by about 0.15. Both GPROF-NN 3D and GPROF-NN HR have slightly higher correlations and GPM CMB achieves the highest correlations with the database precipitation. When compared to MRMS, the correlation of GPROF V7 decreases noticeably making it more comparable to the correlations of GPROF V5. GPROF-NN 1D and GPROF-NN 3D remain fairly close in terms of correlation. GPROF-NN HR yields the highest correlation with the MRMS validation data even surpassing GPM CMB.

In terms of SMAPE, the accuracy of the GPROF-NN HR and GPM CMB retrievals is lower compared to the other retrievals. For GPM CMB, the SMAPE is higher than any of the GPROF-NN retrievals while the SMAPE of the GPROF-NN HR retrieval is higher than that of the GPROF-NN 3D retrieval. The reduced retrieval accuracy of GPM CMB and GPROF-NN HR in terms

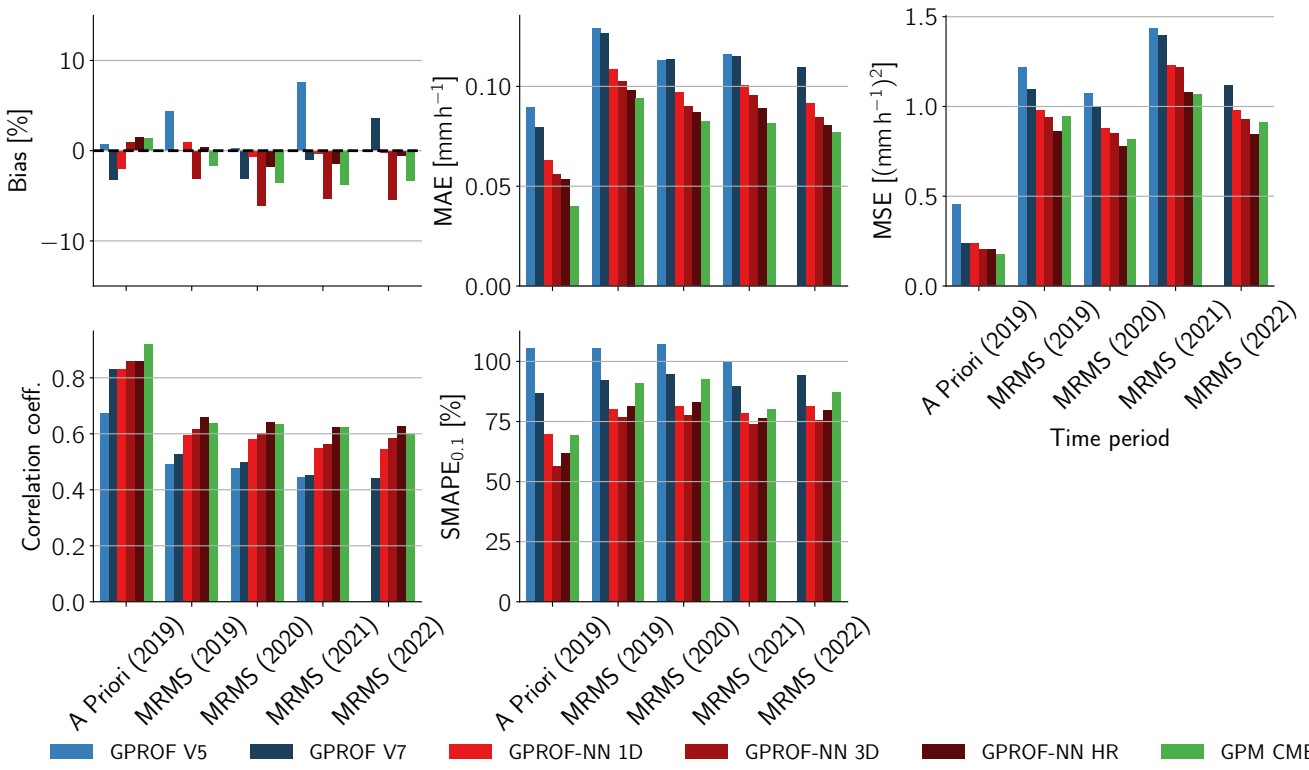

**Figure 5.** Accuracy metrics of the GPROF retrievals compared to the a priori database and MRMS.

of SMAPE may be caused by the SMAPE excessively penalizing the misplacement of precipitation, which may put retrievals with a higher resolution at a disadvantage.

Finally, we evaluate the skills of the GPROF retrievals in detecting precipitation. To this end, precision-recall curves for the detection of precipitation from the a priori database and MRMS validation data are displayed in Fig. 6. The precision of the retrievals is the fraction of truly raining pixels and the total number of pixels retrieved as raining. The recall corresponds to the fraction of actually raining pixels that are detected by each retrieval. Here, *actually raining* is defined with respect to the assumed ground truth, which is the a priori/training database or MRMS validation data depending on which of the two sources

the retrievals are compared against. By varying the detection threshold applied to the retrieved probability of precipitation, a curve of precision and recall values is obtained that fully characterizes the detection skill of each retrieval.

As was to be expected from the previous results, the detection skill with respect to the MRMS validation data is slightly lower than with respect to the a priori/training data. Nonetheless, also for the detection of precipitation consistent improvements between the conventional GPROF retrieval, GPROF-NN 1D, and GPROF-NN 3D can be observed. The skill of GPROF-NN

HR is similar to that of GPROF-NN 3D both in comparison to the a priori database and the MRMS validation data.

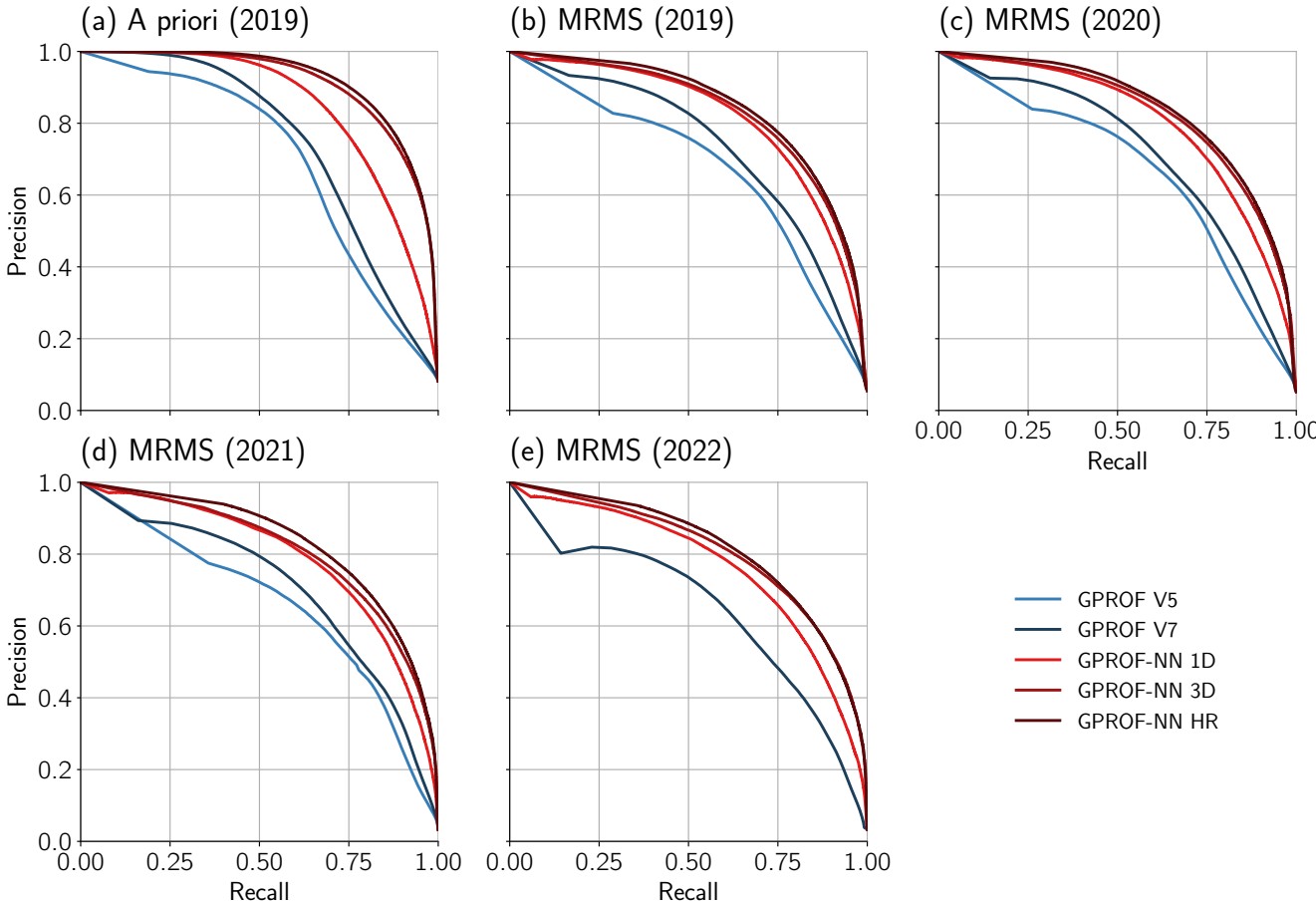

**Figure 6.** Precision-recall curves for the GPROF retrievals. Panel (a) shows the detection skill for the collocations with the a priori/training database. Remaining panels show the results with respect to the MRMS validation measurements for the years 2019, 2020, 2021, and 2022.

### 3.1.2 Regional precipitation characteristics

This section assesses the retrieval accuracy over five regions within CONUS and the K-POL radar in the tropical Pacific in order to assess the ability of the retrievals to represent regional variations in precipitation patterns. The location and extent of the five CONUS regions is displayed in Fig. 7. The five regions encompass the Pacific North-West (NW), the South-West (SW), central CONUS (C), the South-East (SE) and the North-East (NE).

The error metrics for the retrievals in the six regions are shown in Fig. 8. For both the a priori database and the MRMS validation measurements, the regional biases are generally larger in magnitude than they are for the full CONUS. With the exception of GPROF V5, the biases of the GPROF retrievals remain within $\pm 25\%$ indicating that they reproduce the regional climates reasonably well. The biases of the GPM CMB product are of the same sign and similar magnitude as those of the GPROF retrievals, indicating that the biases of GPROF V7 and GPROF-NN are inherited from the a priori/training database.

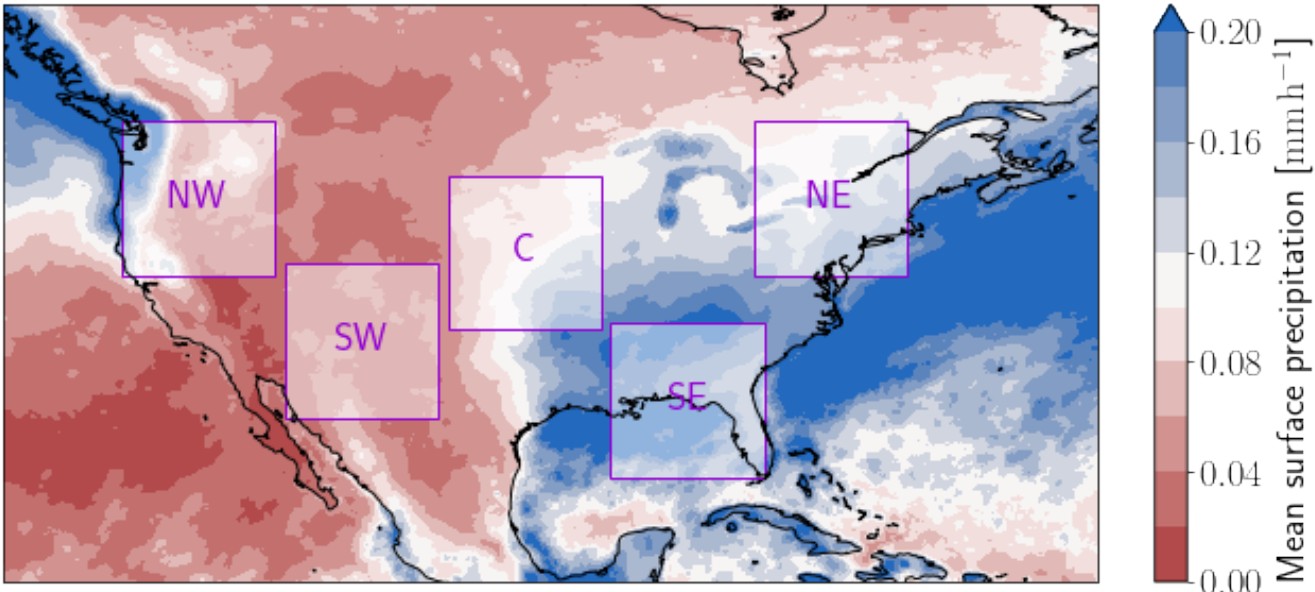

**Figure 7.** IMERG mean precipitation rate over the validation period Oct. 2018 - Sep. 2021. The purple rectangles mark the five regions used for the evaluation of the regional retrieval accuracy.

The biases of GPM CMB over CONUS vary by region with dry biases ocurring in the NW, SW, and SE, wet biases in the C region, and the NE region exhibiting the smallest biases. Temporally, the bias tendencies persist throughout the four assessed years in the eastern and central regions, but are less consistent in the western region especially for the year 2021 and 2022. Except for GPROF V5, the other GPROF retrievals mostly follow the GPM CMB results.

Over the Kwajalein site, the GPM CMB retrieval exhibits considerable wet biases for the years 2020, 2021, and 2022. The GPROF retrievals exhibit comparable biases with the exception of GPROF V5, which yields the smallest biases compared to the validation data. The increase in ocean precipitation for GPROF V7 and the GPROF-NN retrievals can be explained by the inclusion of light precipitation from MIRS where GPM CMB doesn't detect rain and an increase in ocean precipitation in GPM CMB V7 compared to the prevision version of GPM CMB, upon which the GPROF V5 apriori database was based on over ocean surfaces.

Both MAE and MSE vary significantly in magnitude between the different regions with larger errors occurring in regions with more precipitation. Nonetheless, the relative accuracy of the different retrievals is similar to the results from the whole CONUS (Fig. 5): GPROF V7 is slightly more accurate than GPROF V5, GPROF-NN 1D is more accurate than GPROF V7, GPROF-NN 3D more accurate than GPROF-NN 1D, and GPROF-NN HR more accurate than GPROF-NN 3D. This ranking is also reflected in the correlation and SMAPE.

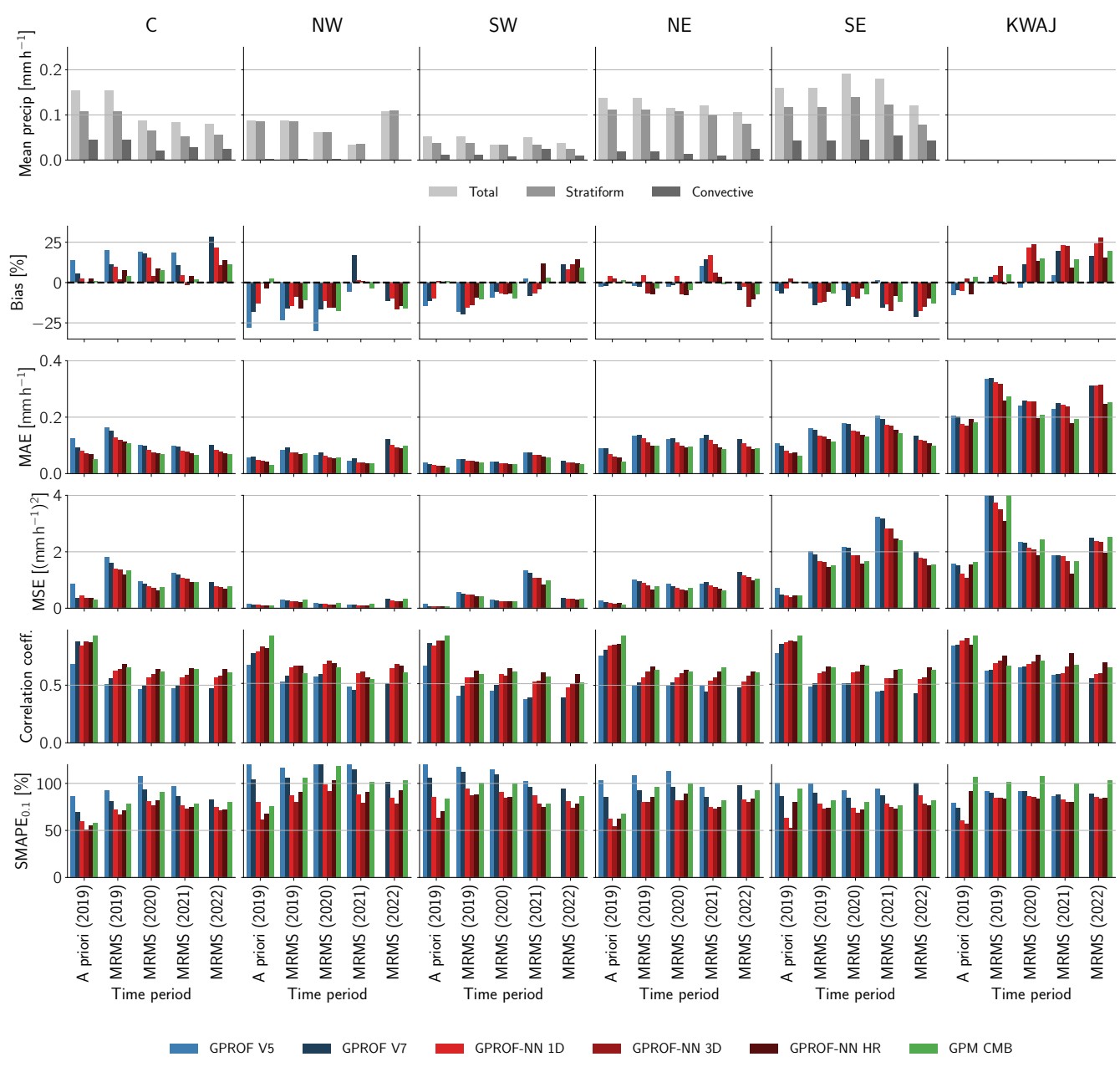

**Figure 8.** Retrieval accuracy metrics for the assessed climate regions over CONUS and the tropical Pacific. The first row shows the mean precipitation rate of the reference data and, where such a classification was available, the contributions from convective and stratiform precipitation. Remaining rows show accuracy metrics for the comparison of the retrieval results against the reference data. Columns from left to right show the results for the for the five considered regions.

Seasonal cycles of precipitation are displayed in Fig. 9. The seasonal cycles were calculated using all collocations from the years 2020 and 2021. Collocations from the year 2019 were excluded because it is part of the retrieval database while the year 2022 was excluded because no GPROF V5 data is available for that year.

GPM CMB and GPROF generally capture the seasonal cycles of the regions accurately. Notable deviations from the MRMS
validation data are an underestimation of stratiform winter precipitation in the North-West as well as an underestimation of convective summer precipitation in the South-East. It should be noted that this analysis excludes snow and precipitation over mountains and snow-covered surfaces and thus the reported biases, especially in the NW region and during winter time, may not be representative of total accumulations. However, even if the overestimation of the excluded cases would make up for this underesimtation, these findings still point towards a notable disagreement between MRMS and the satellite retrievals for
retrievals of liquid precipitation over snow-free and non-mountainous surfaces.

The largest differences between GPROF V5 and V7 are found for warm-season precipitation over the central CONUS, where GPROF V5 exhibits the strongest wet bias, and cold-season precipitation in the NW, where GPROF V5 exhibits a dry bias. Weaker deviations are observed in the South-East, where GPROF V5 captures the warm-season precipitation peak better, and the tropical Pacific, where GPROF V5 retrieves less precipitation during the second half of the year, which puts it closer to the
375 validation measurements.

The results of the GPROF and GPROF-NN retrievals mostly follow those of GPM CMB in most regions. Deviations from GPM-CMB are observed for summer precipitation in the Sout-West. Here, GPM CMB slightly overestimates the MRMS precipitation. Being lower than GPM CMB, the GPROF V7 retrievals agree best with MRMS. The GPROF-NN 1D and 3D retrievals yield results closer to GPM CMB and thus overestimate the MRMS measurements. GPROF-NN HR yields even
higher estimates that GPM CMB thus showing the strongest overestimation compared to MRMS.

Diurnal cycles of precipitation shown in Fig. 10 reveal the diurnal variations of the regional deviations observed above. The underestimation of precipitation in the North-West is found to be related to underestimation of the precipitation during early morning and afternoon. In the South-East, the underestimation is related to convective afternoon precipitation.

The deviations between the GPROF, GPROF-NN and GPM CMB retrievals for summer precipitation in the South-West can
also be observed in the diurnal cycles. Precipitation from GPM CMB, GPROF-NN 3D and GPROF-NN HR peak in the early afternoon, whereas precipitation from GPROF V5 and V7 and GPROF-NN 1D tends to peak later in the afternoon.

The diurnal cycles reveal further differences between the retrievals for convective precipitation in the South-East. Here, GPROF V7 underestimates the convective precipitation most-strongly but also GPROF-NN 1D and 3D retrieve less precipitation than GPM CMB. In contrast to the South-West, the underestimation is more homogeneous across the duration of the
390 precipitation peak.

## 3.2 Effective resolution

For the analysis of the effective resolution of the retrievals, the discrete cosine transform of the retrieved and the MRMS precipitation fields were calculated over non-overlapping windows of size $160\,\mathrm{km} \times 160\,\mathrm{km}$ in which neither any of the GPROF retrievals nor the ground radar measurements had more than $2\%$ of missing values. For this analysis, retrievals over snow-

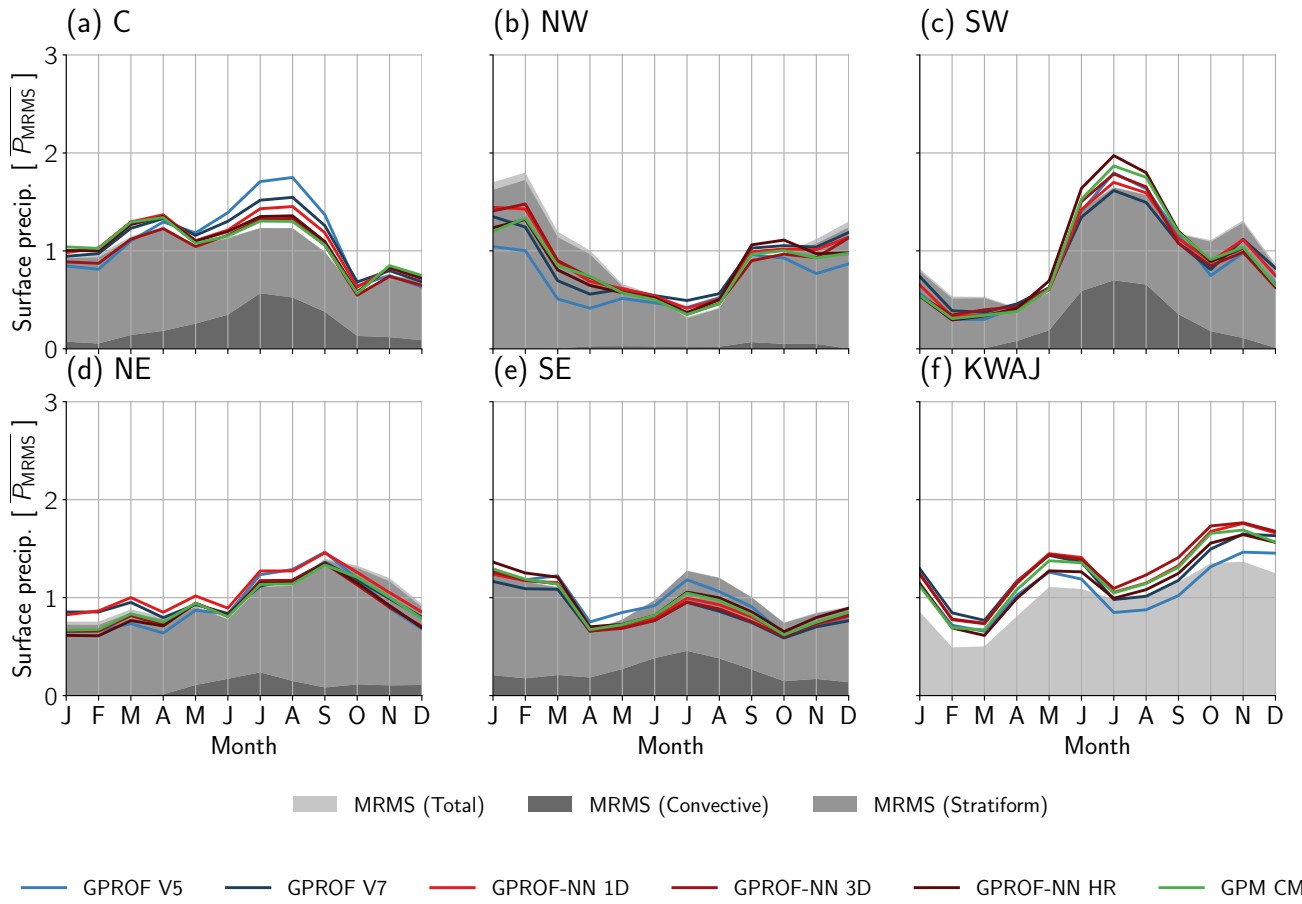

**Figure 9.** Regional seasonal cycles of precipitation. Each panel shows the seasonal cycles derived from GPROF retrievals, GPM CMB and the validation data for one of the considered regions. The seasonal cycles are normalized by the corresponding mean precipitation rate of the validation precipitation. The seasonal cycles were smoothed using a sliding window average with a width of three months. Only collocations from the water years 2020 and 2021 are used.

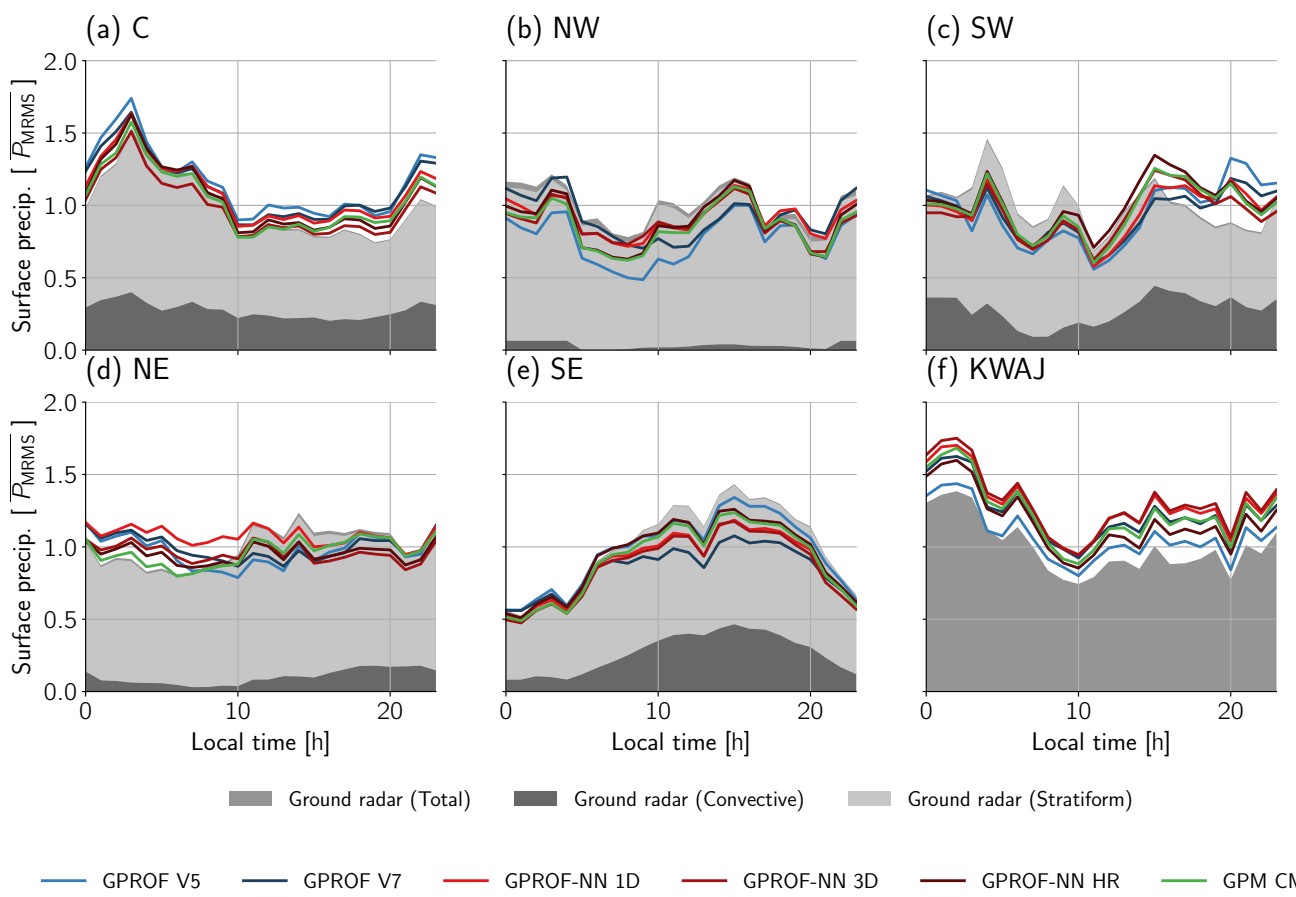

**Figure 10.** Like Fig. 9 but for diurnal cycles of precipitation. The curves were smoothed using sliding window average with a width of 3 hours.

covered and mountains surfaces as well as of frozen precipitation were included in the analysis because omitting these values would have significantly reduced the number of windows with valid precipitation estimates. Of those, however, only windows with a minimum radar-quality index of 0.5 were considered. Only collocations from the water years 2020 and 2021 were used in order to exclude scenes which are also present in the GPROF a priori database. The coherence was computed once using retrievals from the full GMI swath, and once only over the central part of the swath in order to allow comparison to the GPM CMB product.

The spectral coherence between the GPROF retrievals and the reference radar data as a function of the wavelength is displayed in Fig. 11. Over the CONUS, GPROF V7 has slightly higher spectral coherence than GPROF V5. The GPROF-NN retrievals achieve considerably higher coherence than the conventional GPROF retrievals across all wavelengths. The coherence curves of the GPROF-NN retrievals are similar up to spatial scales of about 15 km, above which GPROF-NN 3D achieves

slightly higher coherence than GPROF-NN 1D. The coherence curves of the conventional GPROF retrievals and the GPROF-NN 1D and 3D retrievals are not significantly affected by the swath coverage.

The GPROF-NN HR retrieval achieves the highest coherence of the PMW retrievals, even matching that of the GPM CMB retrieval, when it is evaluated only over the swath center. However, when evaluated over the full swath, the coherence decreases sharply and falls below that of the GPROF-NN 1D retrieval. This indicates that the retrieval overfits on the viewing geometry

of the central part of the GMI swath, which is the only part of the swath for which reference measurements of precipitation are available. Although the training of the GPROF-NN HR retrieval applies data augmentation that simulates the effect of perspective distortion across the swath, it does not account for the parallax effect, i.e., the displacement between observed precipitation signal and the actual precipitation caused by the slant viewing angle of the sensor. Since the parallax effect changes direction towards the edges of the swath, it can potentially explain the decrease in accuracy observed for the GPROF-

NN HR retrieval.

Over the Kwajalein site, all GPROF retrievals yield significantly improved coherence curves due to the direct relationship between microwave signals and the liquid water content in the clouds. Also here, the GPROF-NN 1D and 3D retrievals yield higher coherence than the conventional GPROF retrievals albeit by a smaller margin. The GPROF-NN HR retrieval exhibits similar overfitting as over the CONUS. Evaluated over only the central part of the swath, the coherence curve of the GPROF-

NN HR is higher than that of every other retrieval. Evaluated over the full swath, however, the accuracy decreases below that of the GPROF-NN 1D retrieval. Interestingly, the coherence curve of the GPM CMB retrieval remains at the level of the GPROF-NN HR retrieval evaluated across the full swath and falls below the curves of the GPROF-NN retrievals for scales exceeding $12\,\mathrm{km}$. This is an unexpected result since the GPROF-NN HR retrieval is designed to reproduce the results from the GPM CMB retrieval. A potential explanation could be that the GPM CMB retrieval systematically misses precipitation from shallow

clouds while the PMW retrieval is able to recover some of that precipitation because the signal in the PMW observations is less dependent on the depth of the observed clouds.

The resulting effective resolutions of the retrievals are listed in Tab. 2. The effective resolution for the conventional GPROF retrieval over CONUS are around $45$ and $60\,\mathrm{km}$ for the GPROF V7 and the GPROF V5 retrieval, respectively. The effective resolution of the GPROF V5 retrieval is significantly lower when calculated only over the swath center, but this is caused by the

curve falling slightly below the threshold for the calculation of the effective resolution and does not correspond to a significant difference in the coherence curves. With effective resolution of about $29$ and $21\,\mathrm{km}$, respectively, the GPROF-NN 1D and 3D retrievals significantly improve the effective resolution over the CONUS. The effective resolution also clearly reflects the overfitting behavior of the GPROF-NN HR retrieval, which achieves an effective resolution of $17\,\mathrm{km}$ over the central part of the swath but only $31\,\mathrm{km}$ across the full swath. The effective resolution of the GPM CMB retrieval is $18\,\mathrm{km}$ over the CONUS.

Over the Kwajalein site, the effective resolution of the GPROF V5 retrieval of $16\,\mathrm{km}$ is slightly better than that of the GPROF V7 retrieval with $17\,\mathrm{km}$. Both GPROF-NN retrievals achieve an effective resolution of around $15\,\mathrm{km}$. The effective resolution of the GPROF-NN HR retrieval is $11\,\mathrm{km}$ evaluated at the swath center but decreases to $15\,\mathrm{km}$ over the full swath. The effective resolution of the GPM CMB retrieval is only $15\,\mathrm{km}$.

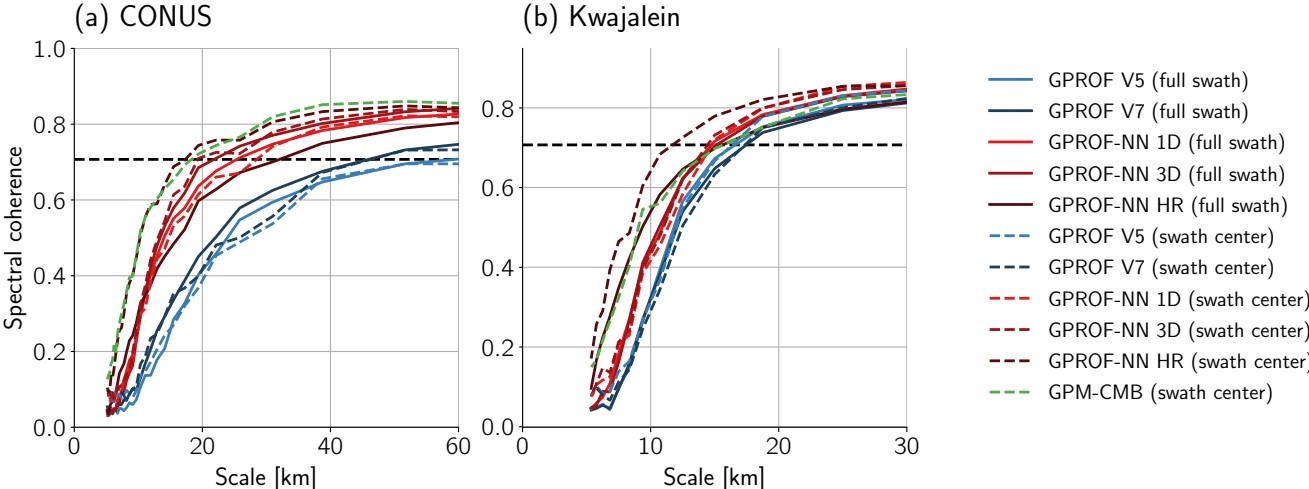

**Figure 11.** Spectral coherence curves for the GPROF retrievals over CONUS (Panel (a)) and the Kwajalein Atoll in the tropical Pacific (Panel (b)). Solid lines show the coherence curve for the evaluation covering the center of the GMI swath. Dashed lines show the coherence curves calculated over the full GMI swath.

| | Effective resolution [km] | | | |
| | CONUS | | Kwajalein | |
| Retrieval | Full swath | Swath center | Full swath | Swath center |
|---|---|---|---|---|
| GPROF V5 | 59.38 | 86.24 | 16.6 | 16.36 |
| GPROF V7 | 45.96 | 46.3 | 17.46 | 17.37 |
| GPROF-NN 1D | 29.49 | 28.48 | 14.75 | 14.77 |
| GPROF-NN 3D | 21.83 | 19.3 | 15.14 | 14.5 |
| GPROF-NN HR | 32.03 | 17.36 | 15.61 | 11.72 |
| GPM CMB | - | 18.32 | - | 15.65 |

**Table 2.** Effective resolution of the GPROF and GPM CMB retrieval evaluated against ground radar measurements.

### 3.3 Retrieval accuracy of GPROF for other GPM sensors

This section evaluates the accuracy of the GPROF and GPROF-NN retrievals for the other sensor types of the GPM constellation for which MRMS validation data is available. One year of collocations with MRMS and K-POL precipitation estimates were extracted for one instance of each sensor type. The corresponding platforms and collocation periods are listed in Table 1. The resulting retrieval accuracy metrics over CONUS and Kwajalein are displayed in Fig. 12 and Fig. 13, respectively.

The biases of the GPROF V7 retrieval are well within $10\%$ over both the CONUS and the Kwajalein site. The exception
are MHS and ATMS, for which the bias for Kwajalein exceeds $30\%$. For most sensors, MSE, MAE and SMAPE are larger than for GMI, which is expected considering that the other sensors of the GPM constellation have lower spatial resolution, fewer channels or both. An exception is TMI, which despite having fewer channels and larger footprints than GMI, achieves higher correlation over CONUS. This is likely due to the reduced latitude coverage of TMI, which only covers the lower part of CONUS and is therefore more likely to observe precipitation with clear signatures in PMW observations.

Compared to GPROF V5, the retrieval biases of V7 tend to be smaller than those of V5. GPROF V7 tends to exhibit larger MAE but reduced MSE and SMAPE. A possible explanation for this may be the treatment of pixels with low probability of precipitation, which were explicitly set to $0.0\,\mathrm{mm\,h^{-1}}$ in GPROF V5. The clearest difference between GPROF V7 and GPROF V5 are observed for SSMIS, for which all accuracy metrics are improved.

The GPROF-NN retrievals exhibit significant spread in biases. Over the CONUS, the retrievals have a tendency to be biased
low with biases reaching up to $20\%$. Over Kwajalein, the retrieval for the conical scanners (TMI, SSMIS, AMSR2) are biased low whereas the retrievals for the cross-track scanners exhibit wet biases of up to $70\%$ for MHS. GPROF-NN 1D yields similar or slightly more accurate results in terms of MAE, MSE and correlation for the conical scanners but yields less accurate results for the assessed cross-track-scanning sensors, MHS and ATMS. For GPROF-NN 3D the improvements are more robust. In terms of the SMAPE, the results are not conclusive with the GPROF-NN retrievals yielding higher errors for some of the sensors.

The results obtained for the GPROF-NN retrievals contrast with the results for GMI, for which very clear improvements over the conventional GPROF retrievals were observed. A likely cause for this is that the training of the GPROF-NN retrievals is based on radiative transfer simulations, which introduce an additional error source into the retrieval. Especially the large biases over the Kwajalein site in the retrievals from the cross-track scanners, whose simulations use a different code than the simulations for the conical scanners, point to problems with the simulations.

### 3.4 Retrievals of frozen precipitation and over snow-covered and mountain surfaces

The analyses of the GPROF retrievals presented in Sec. 3.1 and Sec. 3.3 were limited to precipitation that is not identified as frozen by the MRMS hydrometeor classification and restricted to retrievals over snow-free and non-mountainous surfaces. As mentioned above, this was primarily motivated by both satellite retrievals and ground-based validation measurements being highly uncertain in these scenarios.

Figure 14 displays the relative contribution of the excluded cases to the total precipitation for all validation samples. The samples that the presented analyses focused on accounts for $80\%$ of all precipitation in the MRMS validation data, while

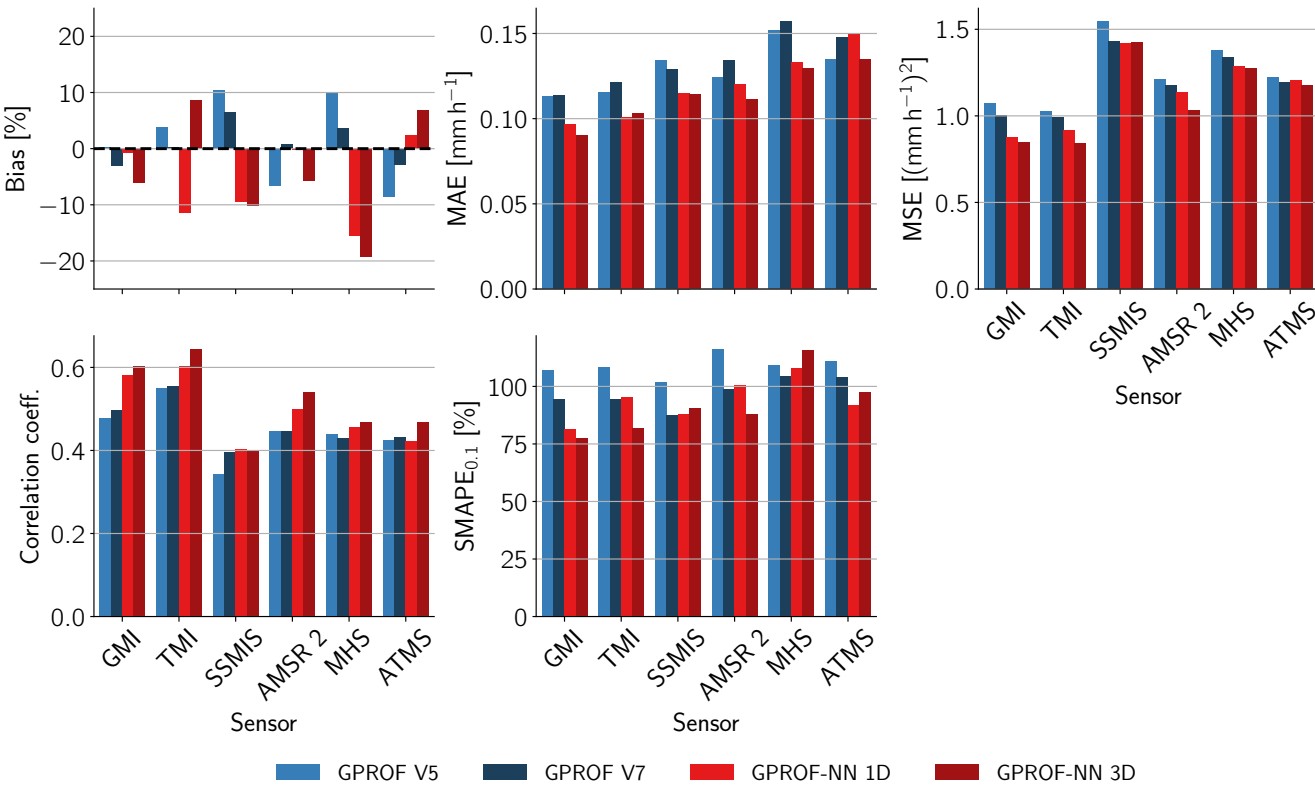

**Figure 12.** Accuracy metrics for a selection of other sensors of the GPM constellation evaluated against gauge corrected MRMS measurements over the CONUS. Results for GMI from the water year 2020 are included for reference.

mountain precipitation accounts for around $15\%$. Snow and precipitation over snow-covered surfaces account for less than $5\%$ of the total precipitation.

GPROF V7, GPROF-NN 1D, and GPROF-NN 3D produce notably higher precipitation over mountain surfaces than MRMS, GPM CMB and GPROF-NN HR. This is because of the orographic enhancement factors that are applied in the a priori/training database of GPROF V7. Since GPROF-NN HR is trained directly on GPM CMB precipitation it does not reproduce this correction for orographic enhancement.

Although GPROF V7, GPROF-NN 1D, and GPROF-NN 3D retrieve more precipitation over mountains, they strongly underestimate snow and weakly underestimate precipitation over snow-covered surfaces. These effects together with the overall smaller contribution of mountain precipitation, cause the relative biases calculated over all validation samples to be similar to those calculated excluding frozen precipitation and retrievals over snow-covered surfaces and mountains.

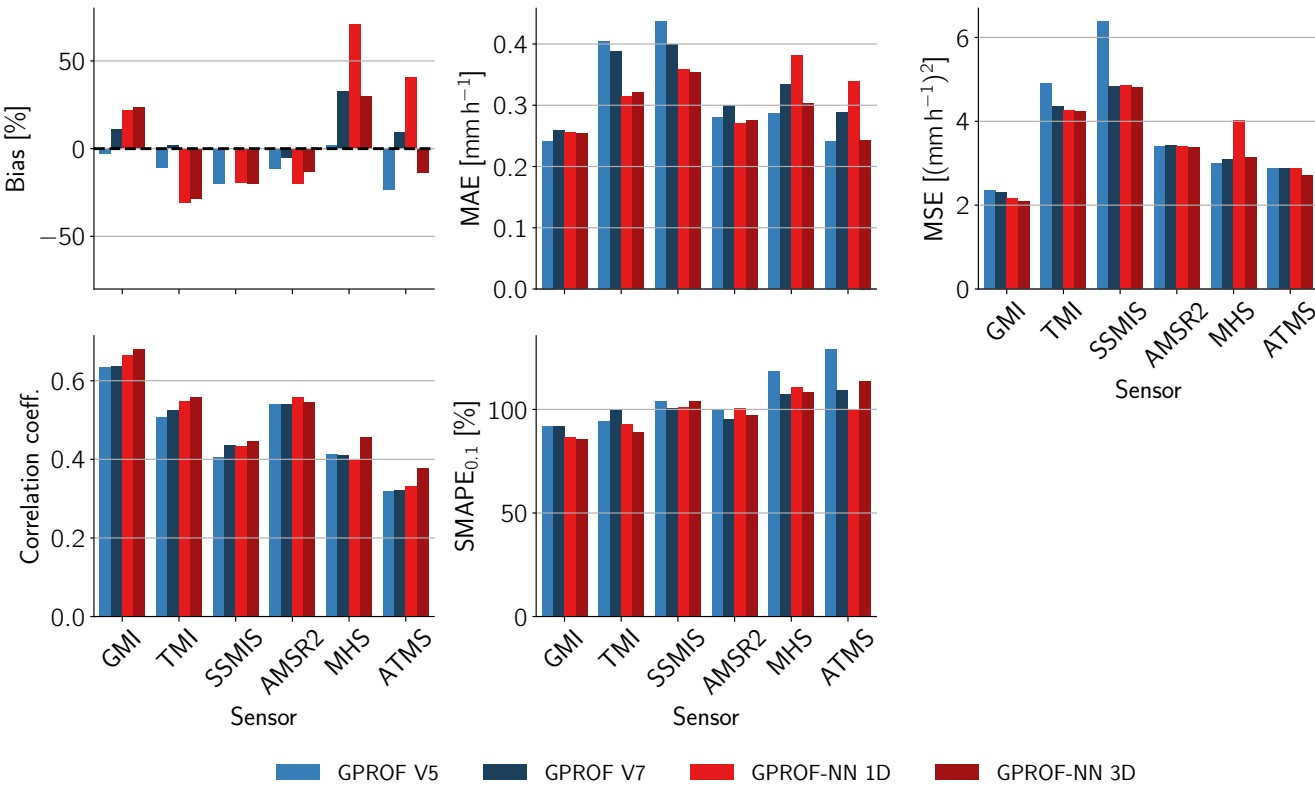

**Figure 13.** Accuracy metrics for a selection of other sensors of the GPM constellation evaluated against the K-POL radar in the tropical Pacific. Results for GMI from the water year 2020 are included for reference.

# 4 Discussion

## 4.1 Limitations of this validation study

Our validation excluded retrievals over snow-covered land surfaces and mountainous terrain because of the enhanced uncer-
tainties in both validation data and retrievals as well as their special treatment in the a priori/training database. This special treatment would skew comparison to the GPM CMB product and thus complicate the attribution of retrieval errors. For similar reasons, we have also excluded precipitation identified as frozen by the MRMS hydrometeor classification.

We have investigated the effect of excluding snow-covered land surfaces, mountainous terrain and frozen precipitation on the retrieval biases over all of the CONUS and found them to robust to these exclusions. The orographic enhancement factors
applied in the a priori database, cause GPROV V7 (and GPROF-NN 1D and 3D) to overestimate MRMS precipitation in mountain regions. In regions where much precipitation occurs in mountain regions, as is the case in the NW region, GPROF

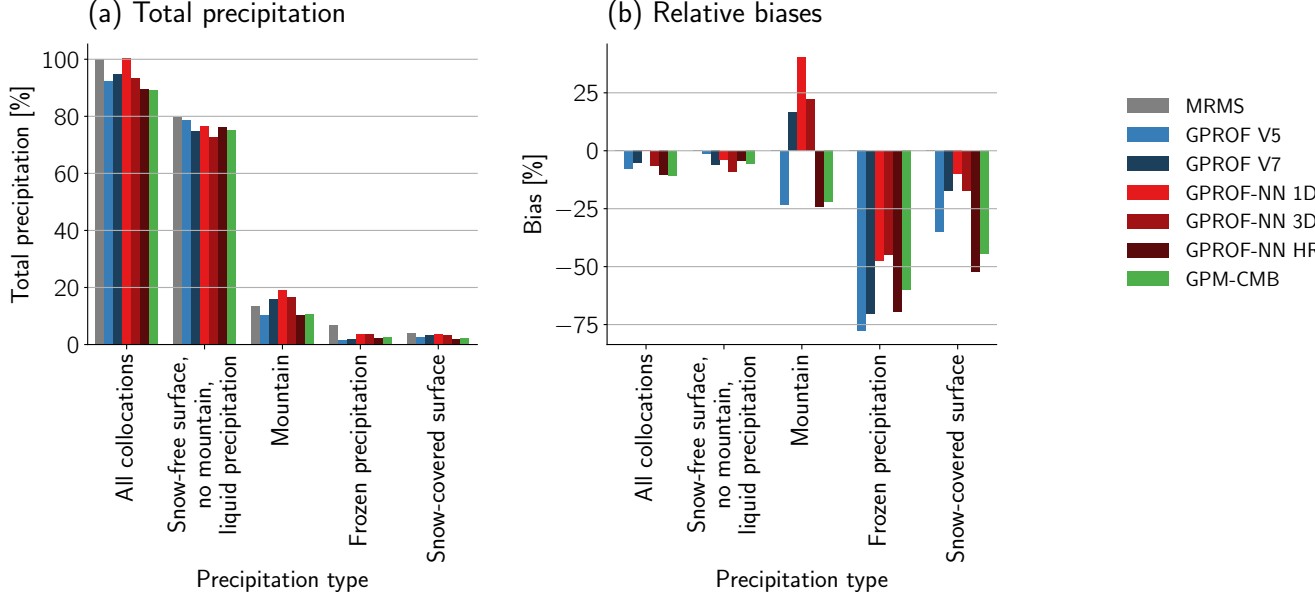

**Figure 14.** Contribution of frozen precipitation and precipitation over snow-covered and mountain surfaces to the total precipitation in the MRMS validation data. Panel (a) displays the total volume of precipitation for different retrieval scenarios relative to the total MRMS precipitation for all validation collocations from the water years 2021 and 2022. Panel (b) shows the corresponding retrieval biases relative to corresponding mean MRMS precipitation.

V7 may thus be able to balance the underestimation found in this analysis. However, when frozen precipitation is included in the accumulations the underestimation of precipitation is more likely to be exacerbated.

Finally, it should be kept in mind that the validation data over CONUS was not sampled uniformly, but is limited to regions where high-quality radar observations are available. Since radar coverage is generally biased towards non-mountainous areas and lower elevations, the contribution of both frozen precipitation and precipitation over mountain surfaces may be underestimated in our analyses.

## 4.2 From GPROF V5 to V7

Two principal differences between GPROF V5 and V7 are the inclusion of light precipitation from MIRS over ocean and the application of orographic enhancement factors, which both act to increase the amount of precipitation in the a priori database. The effect of these changes are reflected in the larger amount of precipitation retrieved by GPROF V7 over the Kwajalein site (Fig. 8) and the higher amount of precipitation retrieved over mountains (Fig. 14). If precipitation over mountain surfaces is included, GPROF V7 retrievals are about $5\%$ wetter over CONUS than those of GPROF V5.

For the retrievals of liquid precipitation over snow-free and non-mountainous surfaces that this study focused on, the retrieval accuracy of GPROF V7 for GMI was similar to the previous version of the algorithm, GPROF V5. While the biases of GPROF

V7 compared to the ground-radar validation data are larger than those of V5, they are closer to GPM CMB, which the retrieval aims to reproduce. Small improvements are found in terms of MAE, MSE, correlation coefficient and SMAPE. The effective resolution of the two retrievals are practically identical.

On regional scales, the most significant changes are a reduction in the overestimation of warm-season precipitation over the central CONUS, reduced underestimation of cold-season precipitation in the North-West, a slightly more pronounced underestimation of warm-season precipitation in the South-East and a more pronounced overestimation of precipitation in the tropical Pacific. Without gauges to calibrate the rainfall product, the Kwajalein site may, however, be more uncertain than the CONUS estimates. For the other sensors of the GPM constellation assessed here, GPROF V7 exhibited smaller biases than GPROF V5 compared to the ground-based validation data.

## 4.3   Attribution of retrieval errors

In order for this validation to inform the future development of GPROF, it is essential to determine whether the observed deviations from the validation data are due to the reference data used by GPROF or the inherent uncertainties of the retrieval and the applied method. Figure 3 revealed a high degree of similarity between biases in GPROF and GPM CMB in both space and time. This indicates that the GPROF retrieval inherits systematic errors from the GPM CMB retrieval. Further evidence for a notable contribution of GPM CMB to systematic errors in GPROF was found in the analysis of regional biases of GPROF, which tend to follow those of GPM CMB (Fig. 8, Fig. 9, Fig. 10).

The attribution of the errors can be formalized by investigating the coefficient of determination between the errors in GPROF and GPM CMB with respect to the validation data. The coefficient of determination represents the fraction of variance of the GPROF errors that can be explained by the GPM CMB errors and thus quantifies the contribution of the GPM CMB errors to the MSE of GPROF. Table 3 lists the coefficients of determination of errors between GPM CMB and GPROF and validation data for instantaneous estimates as well as the annual means at $5°$ resolution shown in Fig. 3. For the GPROF V7 retrievals, the coefficient of determination for instantaneous errors varies between 0.17 and 0.3 over the four assessed years. This means that between 17 and $30\%$ of the mean squared error can be explained by the error in the GPM CMB retrieval. For the GPROF-NN retrievals this rate is consistently higher and reaches up to between 30 and $46\%$ for the GPROF-NN HR retrieval. This indicates that GPM CMB error has a notable influence on the accuracy of GPROF even for instantaneous precipitation estimates.

All rates of explained variability increase drastically when annual accumulations over $5°$ boxes are considered. For GPROF V7 they range from 36 to $67\%$ while they reach between 79 and $93\%$ for GPROF-NN 3D. While the fraction of explained variability in the GPROF V7 biases is around $50\%$ over the 4-year period, it increases to about $75\%$ for GPROF-NN 1D, $80\%$ for GPROF-NN HR and $85\%$ for GPROF-NN 3D. This confirms that systematic regional errors in the GPM CMB data contribute significantly to regional biases in GPROF.

The role of GPM CMB for the systematic error in GPROF becomes even clearer when the regional biases of GPM CMB are subtracted from the biases of GPROF. The resulting maps of reference-data-corrected GPROF biases are shown in Fig. 15. Subtracting the GPM CMB biases reduces the overall magnitude of the biases and their temporal and spatial variability. The

| | Instantaneous | | | | 5° annual mean | | | |
|---|---|---|---|---|---|---|---|---|
| Retrieval | 2019 | 2020 | 2021 | 2022 | 2019 | 2020 | 2021 | 2022 |
| GPROF V7 | 0.17 | 0.19 | 0.30 | 0.30 | 0.36 | 0.37 | 0.67 | 0.67 |
| GPROF-NN 1D | 0.20 | 0.25 | 0.34 | 0.37 | 0.76 | 0.73 | 0.69 | 0.79 |
| GPROF-NN 3D | 0.22 | 0.26 | 0.36 | 0.39 | 0.79 | 0.87 | 0.84 | 0.93 |
| GPROF-NN HR | 0.31 | 0.33 | 0.42 | 0.46 | 0.88 | 0.85 | 0.52 | 0.95 |

**Table 3.** Coefficient of determination between errors in GPM CMB and GPROF retrieval errors.

reduction in bias is more pronounced for the GPROF-NN retrievals indicating that their higher accuracy puts increased weight
on the reference data.

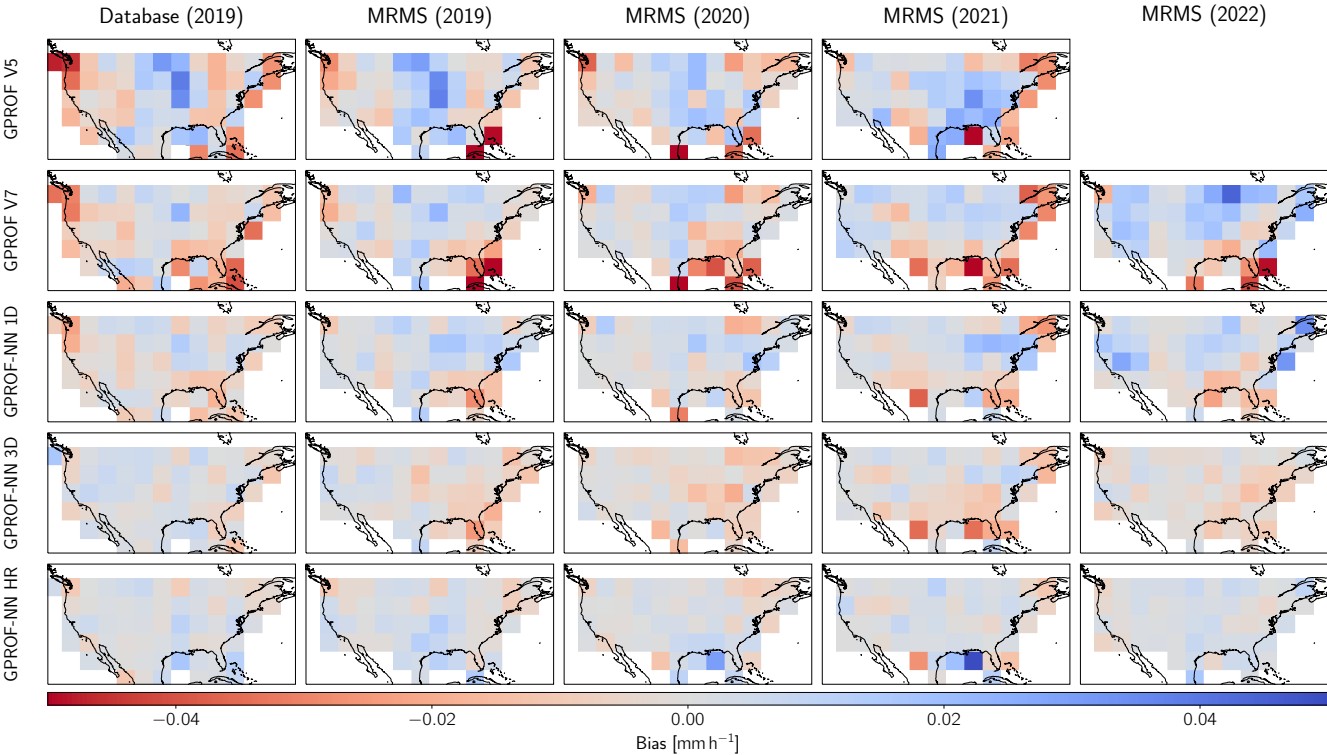

**Figure 15.** Spatial distribution of retrieval biases corrected for biases in GPM CMB. Each panel shows the biases calculated over a uniform
longitude-latitude grid with a grid size of 5°. The corresponding biases of GPM CMB have been subtracted from each bias field. Rows show
the results for the different retrieval algorithms. Columns show the results for different reference data and time periods.

## 4.4 Validation of the GPROF-NN retrievals

The assessment of the GPROF-NN retrievals against ground-radar measurements confirmed the significantly higher retrieval accuracy of the neural-network based implementation for GMI. Although the biases of GPROF-NN 1D are similar to those of GPROF V7, consistent improvements of the order of $10 - 20\%$ are observed in all other assessed accuracy metrics. Significant improvements were also found in detection of precipitation and the effective resolution of the retrieval. The effective resolution of the GPROF-NN retrievals over land is more than twice as high as that of GPROF V7. This is a very encouraging result, considering that these improvements are comparable to those between sensor generations. The fact that a simple algorithm update can unlock these improvements, highlights the importance of improving the retrieval methodology for current and historical space-borne precipitation estimates.

Even higher accuracy is achieved by GPROF-NN 3D and the GPROF-NN HR retrievals. However, the analysis of the effective resolution presented in Sec. 3.2 indicates that the retrieval is overfitting to the viewing geometry at the center of the swath. It seems likely that the decrease in retrieval accuracy towards the edges of the swath is due to the parallax effect, which changes direction towards the edges of the swath. Since reference precipitation measurements are only available at the center of the swath, the neural network cannot learn that the direction of the parallax effect changes across the swath. Fully realizing the potential of the GMI sensor may therefore require incorporating this effect into the training process. Furthermore, the overfitting of the GPROF-NN 3D and GPROF-NN HR retrievals points to a general limitation of using data based on the GPM DPR to train and evaluate GMI retrievals that are based on convolutional neural networks. When only data from the center of the swath is used to train and evaluate a retrieval, its accuracy will likely be overestimated.

Nonetheless, the results of the GPROF-NN 3D and especially the GPROF-NN HR retrievals are worth highlighting. The GPROF-NN HR retrieval demonstrates that precipitation retrievals from GMI can achieve retrieval accuracy that is on par with that of the GPM CMB retrievals and significantly higher than that of the current GPROF retrievals for GMI. The analysis of the contribution of the GPM CMB retrieval errors indicates that with a suitable neural-network-based retrieval systematic errors at regional scales are limited by the accuracy of the reference data which accounts to up to $37\%$ of the MSE of instantaneous precipitation estimates and up to $95\%$ of the variability in annual accumulations at regional scales.

Unfortunately the improvements observed for GMI did not carry over to the other sensors of the constellation. The principal difference for those sensors is that the a priori database uses simulated brightness temperatures instead of real observations. Errors in the simulated brightness temperatures thus constitute an additional error source that incurs a generalization error when the retrieval is applied to real observations. Pfreundschuh et al. (2022) found clear improvements in retrieval accuracy of the GPROF-NN retrieval also for the MHS sensor when the retrieval is evaluated against the a priori database. This indicates that the simulation errors are to blame for the reduced accuracy of the GPROF-NN retrievals for the other GPM sensors. Nonetheless, the GPROF-NN 3D retrieval improves most error metrics compared to the GPROF V7 retrieval. This indicates that the incorporation of structural information helps to make the retrieval more robust.

The largest retrieval biases were found for the GPROF-NN 1D retrieval for the cross-track scanners over the Kwajalein site where wet biases up to $70\%$ were found. This may indicate a particular problem with the simulation of the observations from

575 cross-track scanners over ocean. While not as extreme, the GPROF-NN retrievals exhibit higher biases than the GPROF V7 retrieval over CONUS.

It should be noted, however, that the same simulations are used by GPROF V7, which appears to be less affected by the suspected simulation errors. The improved robustness of the Bayesian scheme used by GPROF V7 is likely due to the hard-coded channel uncertainties, which take into account the forward simulation error of each sensor. In contrast to that, the
580 neural-network retrievals learn the observation uncertainties from the training data.

Another notable difference between the GPROF and the GPROF-NN retrievals is that GPROF applies an additional calibration of the precipitation that ensures that the annual mean precipitation from other sensors matches that of GMI. Such a calibration has not been included in the GPROF-NN retrievals. Furthermore, it should be noted that the measurements from the K-POL radar can be as much as $20\%$ lower than the GPM CMB retrievals, which highlights the difficulty of finding reliable
validation measurements over the ocean.

## 4.5 Future improvements of GPROF

The results presented in this study show that observations from GMI can be used to produce precipitation estimates at essentially the same accuracy as the GPM CMB retrievals with respect to gauge-corrected ground-based radar measurements. However, achieving this level of accuracy across the full swath will require including reference data from the outer edges of the
590 GMI swath in the training data. In principle, this can be achieved through synthetic transformations of the existing training data that account for the parallax effect. However, a technically simpler approach is including reference data from other sources in the training. Given that these improvements can be retroactively applied to the full record of GMI observations, we consider this worthy of further development work.

The largest impediment to an operational adoption of the GPROF-NN retrievals is the retrieval bias observed for other
sensors. The reason for this are likely errors in the assumed hydrometeor profiles and/or radiative transfer simulations used to generate the a priori databases for these sensors. We thus identify the radiative transfer simulations used for the GPROF a priori database as a critical limitation for the accuracy of GPM precipitation estimates and suggest that future research should systematically assess and improve these simulations. Alternatively, collocations between GPM CMB and the other sensors may be incorporated in the training process to improve the retrievals from these sensors.

However, our results also show that systematic retrieval errors of all GPROF retrievals largely follow the biases of the GPM CMB retrieval. In the longer term, improvements of global precipitation measurements will therefore require improvements in what are today considered reference measurements of precipitation, i.e. (potentially gauge-corrected) ground-radar measurements and space-borne combined radar/PMW radiometer measurements. Convergence of these measurements is a prerequisite to better constrain global precipitation measurements from meteorological satellites.

## 5 Summary and conclusions

This study validated the most recent version of the GPROF PMW precipitation retrieval, GPROF V7, its predecessor, GPROF V5, and the experimental GPROF-NN retrievals, which are candidate algorithms for the next version of GPROF. The assessment was based on multiple years of collocations with gauge-corrected ground radar QPEs over CONUS as well as non-gauge-corrected ground-radar QPEs from the Kwajalein atoll. The validation data included collocations between the GPROF retrieval database and the ground-based validation data as well as the GPM CMB product and the validation data in order to separate the contributions of the reference data used by the GPROF retrieval and the retrieval method.

The validation of GPROF V7 surface precipitation revealed the following characteristics:

1. GPROF V7 slightly underestimates liquid precipitation over snow-free and non-mountainous surfaces over CONUS with dry biases varying between 2 and $8\%$ during the 4 surveyed years. At a resolution of $5\,\mathrm{km}$, retrieval errors range from $0.11$ to $0.13\ \mathrm{mm\,h^{-1}}$ in terms of MAE, $1.1$ and $1.6\,(\mathrm{mm\,h^{-1}})^2$ in terms of MSE, and 80 to $90\%$ in terms of $\mathrm{SMAPE}_{0.1}$. The correlation ranges from $0.45$ to $0.55$.

2. On regional scales, consistent dry biases were observed in the South-East, where annual biases reached up to $20\%$. Over the central CONUS consistent wet biases between 2 and $25\%$ were observed. Over the other regions, biases varied between $-20\%$ and $+20\%$.

3. Over the tropical Pacific GPROF V7 GMI exhibits a wet bias ranging from 2 to $18\%$ compared to measurements from the K-POL radar. However, the retrieval remains within $10\%$ of the GPM CMB product.

4. GPROF V7 GMI systematically underestimates liquid cold-season precipitation over snow-free and non-mountainous surfaces in the North-West and warm-season, afternoon convective precipitation in the South-East.

5. Annual biases at $5°$ resolution are moderately correlated with GPM CMB indicating that differences between GPM CMB and the ground-radar measurements contribute to the observed biases.

6. GPROF V7 GMI is about $5\%$ dryer than GPROF V5 over snow-free non-mountainous surfaces over CONUS but significantly wetter over mountain surfaces leading to about $5\%$ wetter retrievals for all validation samples. Over tropical oceans the inclusion of MIRS precipitation and increases in GPM CMB V7 precipitation make GPROF V7 retrievals about $10-20\%$ wetter over the tropical Pacific. Comparable tendencies were observed for TMI, SSMIS, AMSR2, MHS and ATMS. MAE, MSE, SMAPE and correlation were found to be mostly identical.

Based on these findings, we conclude that GPROF V7 reliably captures principal precipitation characteristics from continental and regional scales. However, significant spatial and temporal variability in the retrieval errors make it difficult to generalize the validation results to a global context.

Furthermore, our results confirm the potential of the neural-network-based implementations of GPROF to improve the GPM PMW retrievals. Consistent improvements in the accuracy of instantaneous precipitation estimates are observed for GMI. In

particular, we find a doubling of the effective resolution of the retrievals over land. Even larger improvements can be achieved through the use of retrievals that exploit spatial patterns in the input observations. In particular, we have shown that a GMI super-resolution retrieval can achieve similar retrieval accuracy and effective resolution as the GPM CMB product. However, these were found to overfit on the GMI viewing geometry at the center of the swath and will thus require further development before they can achieve high accuracy across the full swath. In addition to that, the accuracy of the hydrometeor profiles and/or radiative transfer simulations of the GPROF a priori database were identified as a principal impediment for unlocking the improvements found for GMI for the other sensors of the GPM constellation. The presented results demonstrate that the current observational record of precipitation could be improved significantly by adopting the neural-network based retrieval for the operational processing of the observations from the GPM constellation but that further development is needed to harness the full potential of the PMW observations from the GPM constellation.

*Code and data availability.* The implementation of all GPROF-NN retrievals as well as the code used to extract the validation collocations analyze the results is available from Pfreundschuh (2022). GPROF level 2A retrieval results are available from GPM Science Team (2022c, b, d, e, a). GPM CMB retrieval are available from Olson (2022). The gauge-corrected MRMS measurements for overpasses of GPM sensors over CONUS as well as the measurements from the K-POL radar are available from Wolff (2023). The validation collocations created for this study are not publicly available due to the significant storage requirements. We are, however, happy to share them upon request.

## Appendix A: Quantitative accuracy metrics used in this study

Ranking the quality of precipitation estimates is a non-trivial problem because what constitutes a good estimate depends heavily on the downstream application. The ultimate motivation for this work is to improve the precipitation estimates from the PMW sensors of the GPM constellation in a way that benefits all possible downstream applications. To ensure that we are working towards an actual improvement of these estimates instead of simply tuning the results to improve a single error metric, we use a selection of error metrics to evaluate the retrievals.

The quantitative error metrics that we use in this study are listed together with their formulas and valid range in table A1. The behavior of the error functions of mean squared error (MSE), mean absolute error (MAE), and symmyetric mean percentage error (SMAPE) is illustrated in Fig. A1. Since both MSE and MAE depend on directly on the absolute difference between estimate and reference value, the largest errors occur in the regions where either the estimate or the reference precipitation is heavy. This effect is exacerbated by the quadratic nature of the MSE.

Relative errors such as MAPE and SMAPE increase sensitivity to deviation at light precipitation rates by normalizing the error. However, since the MAPE uses only the reference precipitation for normalization it is unsymmetric and will thus favor estimates that underestimate the reference value. Since this would bias the evaluation towards retrievals that underestimate precipitation, we use the SMAPE in this study, which uses a symmetric normalization term.

The MSE, MAE, and SMAPE are evaulated by calculating their sample mean over all validation samples. Their final values are thus the combined result of the error function and the joint occurrence of retrieved and reference precipitation values.

**Table A1.** Quantitative accuracy metrics used in this study. We use over bar to denote the sample mean and $\sigma$ the sample standard deviation taken over all valid measurements.

| Name | Formula | Lower bound | Upper bound | Optimal value |
|---|---|---|---|---|
| Bias | $\overline{P_{\text{Retrieved}} - P_{\text{True}}}$ | $-\infty$ | $+\infty$ | 0 |
| Mean squared error (MSE) | $\overline{(P_{\text{Retrieved}} - P_{\text{True}})^2}$ | 0 | $\infty$ | 0 |
| Mean absolute error (MAE) | $\overline{|P_{\text{Retrieved}} - P_{\text{True}}|}$ | 0 | $\infty$ | 0 |
| Symmetric mean absolute percentage error with threshold $t$ (SMAPE$_t$) | $\overline{\left( \frac{|P_{\text{Retrieved}} - P_{\text{True}}|}{\frac{1}{2}(|P_{\text{Retrieved}} + P_{\text{True}}|)} \right)}$ calculated only over samples with $P_{\text{True}} \geq t$ | 0 | 200% | 0 |
| Correlation coefficient | $\overline{\frac{(P_{\text{Retrieved}} - \overline{P_{\text{Retrieved}}})(P_{\text{True}} - \overline{P_{\text{True}}})}{\sigma(P_{\text{Retrieved}})\sigma(P_{\text{True}})}}$ | 0 | 1 | 1 |

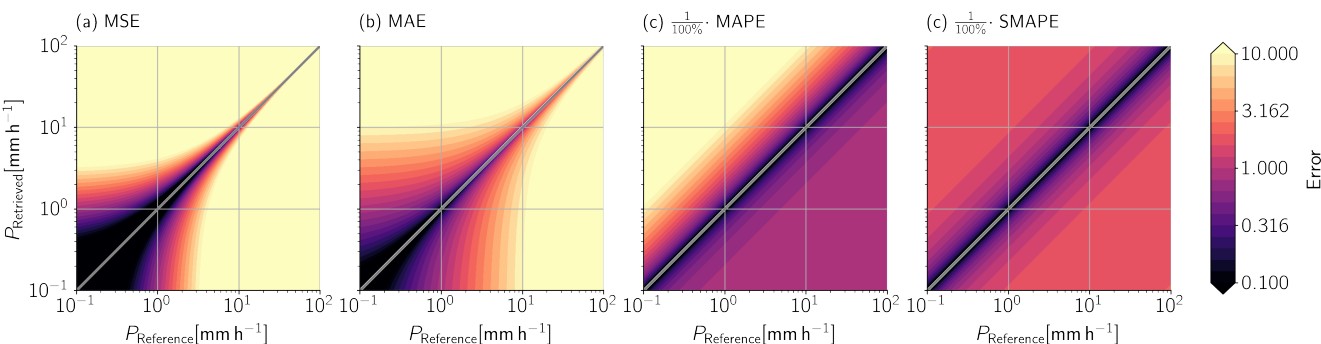

**Figure A1.** Error functions for evaluating precipitation retrievals. Panel (a) shows the value of the MSE for different combinations of retrieved and reference values. Panel (b), (c), and (d) show the according behavior for the MAE, MAPE, and SMAPE.

To illustrate the different characteristics of the three error metrics, Fig. A2 shows the relative contributions from different reference precipitation intensities to the error metrics calculated using all collocations between MRMS and GPROF V7 in the
670 water years 2021 and 2022. As the three curves show, the three error metrics have very different contribution profiles across the spectrum of reference precipitation. While the SMAPE is most sensitive to errors at light precipitation, the MAE has a fairly flat contribution profile with a peak at moderate precipitation, and the MSE is dominated by errors at heavy reference precipitation values.

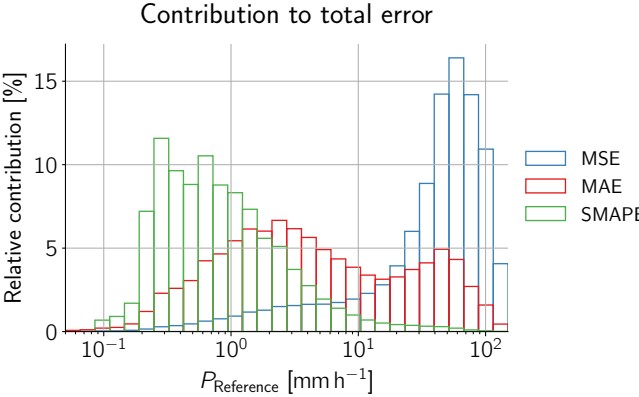

**Figure A2.** Relative contribution of different precipitation rates to total value of each error metrics. The three bar plots show the relative contribution from each corresponding bin the total value of each error statistic. The contributions were calculated for the GPROV V7 retrieval using all validation samples from the water years 2020 and 2021.

*Author contributions.* SP has designed and implemented the study and lead the writing of the manuscript. CG has evaluated the effective
resolution of the retrievals and contributed to writing the manuscript. PB helped gathering the data required for the validation. All authors
have contributed to the writing process through discussion and feedback.

*Competing interests.* No competing interests are present.

*Acknowledgements.* We would like to acknowledge Pierre Kirstetter for the provision of the MRMS radar measurements for GPM ground
validation and advice on the use of the data. Furthermore, we would like to acknowledge David B. Wolff and Jason L. Pippit for providing
guidance on the use of the data from the K-POL radar.

The computations for this study were performed using several freely available programming languages and software packages, most
prominently the Python language (The Python Language Foundation, 2018), the IPython computing environment (Perez and Granger, 2007),
the numpy package for numerical computing (van der Walt et al., 2011), xarray (Hoyer and Hamman, 2017) and satpy (Raspaud et al., 2021)
for the processing of satellite data and matplotlib (Hunter, 2007) and cartopy (Met Office, 2010 - 2015) for generating figures.
Finally, we would like to thank the two anonymous reviewers to provide valuable feedback on the first version of this manuscript.

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
