# Peer review of "GPROF V7 and beyond: Assessment of current and potential future versions of the GPROF passive microwave precipitation retrievals against ground radar measurements over the continental US and the Pacific Ocean"

_EGUsphere, 2023_

## Author Comment (AC1)

**GPROF V7 and beyond: Assessment of current and potential future versions of the GPROF passive microwave precipitation retrievals against ground radar measurements over the continental US and the Pacific Ocean**

Response to reviewers

**1 Response to reviewer 1**

We thank the reviewer for investing their time to read our manuscript and provide constructive feedback to improve it. The principal changes that we have implemented based on comments from both reviewers are the following:

1. We repeated our analysis excluding all precipitation classified as frozen by MRMS, i.e., excluding not only snow but also hail. Significant effects of excluding hail were only observed in the SW region and we have updated the manuscript accordingly.

2. We have added a short section that analyzes the impact of excluding both frozen precipitation as well as precipitation of snow-covered and mountain surfaces and the behavior of the different retrievals over those surfaces.

3. We have revised all references to the a priori database and the MRMS validation data to make the distinction of the two clearer.

Finally, we have also updated that precision-recall curves to show the precision over the recall instead of the recall over the precision, which is the more common way to present these curves. During this, we also realized that the PR curves previously included samples over ocean, which we have now excluded. This lead to a minor change in the relative skill of the GPROF-NN HR retrieval.

In what follows, line and figure numbers are given with respect to the revised manuscript.

**1.1 Principal comments**

**Reviewer comment 1**

The whole validation is based on 'validation measurements' and 'reference measurements', them being ground-based radars and GPM CMB precipitation retrievals respectively. In section 3.1.1 sometimes there is a bit of confusion on how the different databases are addressed. Line 226 has 'reference precipitation', line 234 has 'retrieval database', line 270 has 'database', line 281 has 'a-priori database', line 284 has 'database precipitation' etc. I suggest to get a bit of consistency since the whole validation is based on different but very similar databases (GPM CMB 2019 is the a-priori database, GPM CMB for other years is just a comparison database etc.).

**Author response:**

We thank the reviewer for pointing this out. Although we did make an effort to apply language that makes the different between the two datasets clear already in the initial

version of the manuscript, we apparently fell short of this goal. We have revised the references to the two datasets in section 3.1.1 and the rest of the manuscript in an effort to make the distinction clearer.

**Changes in manuscript:**

1. We have revised Sec. 3.1 and the rest of the manuscript taking care to always clearly state whether we are referring to the a priori database or the MRMS validation data.

2. We have revised all figures to consistently use 'A priori' or 'MRMS' to refer to data from these two sources.

**Reviewer comment 2**

In section 3.1.2 you compare different regions and some of the explanations for high biases are attributed to winter precipitation (see line 331 for example). I am a bit confused on how you are dealing with winter precipitation since snow covered surfaces and MRMS frozen precipitation are excluded from the analysis. Please provide more context on how you analyze winter precipitation in the different regions.

**Author response:**

Since our analysis excludes all samples over snow-covered and mountain surfaces as well as precipitation identified as frozen according to MRMS also our analysis of winter precipitation only applies to liquid precipitation over snow-free, non-mountain surfaces.

Although, in general, the contribution of the exluded samples to the total precipitation is rather small, it can have a significant impact on seasonal and regional scales, as is certainly the case for the winter in the NW region. Nonetheless, even if the excluded precipitation estimates can make up for the observed underestimation, our analysis still points to differences between the retrievals and the MRMS validation data.

However, we agree with the reviewer that this must be clearly communicated and discussed in the manuscript. We therefore revised the discussion of the regional seasonal cycles and also added a sub-section that discusses the behavior of the retrieval for different surface and hydrometeor types. Furthermore, we revised relevant statements in the abstract and conclusions to clearly state that our validation focused on liquid precipitation over these surfaces.

**Changes in manuscript:**

1. We have extended the paragraph discussing the regional seasonal precipitation cycles:

   **Changes starting in line 364:**

   GPM CMB and GPROF generally capture the seasonal cycles of the regions accurately. Notable deviations from the MRMS  validation data are an underestimation of stratiform winter precipitation in the North-West as well as an underestimation of convective summer precipitation in the South-East. It should be noted that this analysis excludes snow and precipitation over mountains and snow-covered surfaces and thus the reported biases, especially in the NW region and during winter time, may not be representative of total accumulations. However, even if the overestimation of the excluded cases would make up for this underesimtation, these findings still point towards a notable disagreement between MRMS and the satellite retrievals for retrievals of liquid precipitation over snow-free and non-mountainous surfaces.

2. We added the information that our validation focuses on liquid precipitation over snow-free and non-mountainous surfaces to the abstract and the conclusions.

3. We have included a new section that analyzes the general retrieval characteristics for the excluded regimes and analyzes the impact of excluding those samples. This section includes the new figure shown in Fig. 1.1.

4. We have extended the discussion of the limitation of our study, which now also discusses the potential effect of the left-out samples as well as the potential effect the spatial sampling of the validation data in this.

**1.2 Other suggestions**

**Reviewer comment 1**

Line 37: 'resolution of 10 km' - given the global nature of IMERG maybe 0.1x0.1deg is more appropriate?

**Author response:**

This is correct. We will change this in the manuscript

**Changes is manuscript:**

**Changes starting in line 38:**

.... satellites and rain gauges to produce the level 3 GPM IMERG product (Huffman et al., 2020), which provides global precipitation estimates at a nominal spatial resolution of  0.1° and a temporal resolution of 30 min.

**Reviewer comment 2**

Line 171: 'neighboring pixels' - is this the distance from the centers of neighboring pixels?

[Figure]

Figure 1.1: Contribution of frozen precipitation and precipitation over snow-covered and mountain surfaces to the total precipitation in the MRMS validation data. Panel (a) displays the total volume of precipitation for different retrieval scenarios relative to the total MRMS precipitation for all validation collocations from the water years 2021 and 2022. Panel (b) shows the corresponding retrieval biases relative to corresponding mean MRMS precipitation.

**Author response:**

Yes, that is the distance between pixel centers.

**Changes in manuscript:**

We have included this information in the revised manuscript.

> **Changes starting in line 184:**
>
> The distance between the centers of neighboring pixels of the GMI swath is approximately 13.5 km in along-track direction and 5 km in across-track directions at the center of the swath.

**Reviewer comment 3**

Line 217: 'conditioned on the validation precipitation' - do you mean the analysis is made only on pixels where it is precipitating according to the validation (MRMS) dataset?

**Reviewer comment 4**

Scatter plots over log-scales are quite naturally limited to precipitating pixels since they only display the behavior of the retrieval for non-zero a priori or MRMS precipitation

rates. However, the rest of our analysis always includes both raining and non-raining pixels.

To make this point clearer, we reformulate the sentence in question to stress that the conditioning refers to the distributions displayed in Fig. 2 and not the conditioning of the remaining analyses on raining pixels.

**Changes in manuscript:**

> **Changes starting in line 236:**
>
>  The conditional distributions of retrieved instantaneous precipitation conditioned on the  corresponding precipitation from the a priori/training database and validation measurements from MRMS are displayed in Fig 2.

**Reviewer comment 5**

Line 218: 'GPROF a priori database' - since this database is the same as a priori or training for NN, maybe use 'a priori/training'.

**Author response:**

We have adopted the suggestion in the revised version of the manuscript.

**Changes in manuscript:**

> **Changes starting in line 237:**
>
> The first column of panels shows the distribution of retrieved precipitation with respect to the GPROF a priori/training database.

**Reviewer comment 6**

Line 221: can the spread also be due to the preprocessing clustering?

**Author response:**

Our analysis uses the un-clustered database in order to avoid potential side effects that the clustering could have.

We would like to thank the reviewer for bringing up this point, as it should be made clear in the description of the validation data. We will modify the manuscript accordingly.

**Changes is manuscript**

**Changes starting in line 134:**

The GPROF V5 and V7 retrievals cluster the a priori database based on the similarity of the observations in order to reduce the computational complexity of performing the retrieval. However, both the training of the GPROF-NN retrievals as well as the analysis presented here use the unclustered database.

**Reviewer comment 7**

Line 226: 'conditioned on the reference precipitation' - is this the same as line 217, 'validation precipitation'? As mentioned in comment 1, there is a bit of confusion in the naming of the different datasets used.

**Author response:**

We have revised this sentence to clearly state that this is with respect to the MRMS validation precipitation.

**Changes in manuscript:**

**Changes starting in line 251:**

Apart from this, however, the accuracy of each retrieval conditioned on the MRMS validation precipitation exhibits little inter-annual variability.

**Reviewer comment 8**

Line 229: 'GPROF V5 is based on a different a priori database' - I suggest to specify that V5 was based on DPR over land and CMB over ocean.

**Author response:**

That is a very good point, we will include this in the manuscript.

**Changes in manuscript:**

**Changes starting in line 255:**

Since GPROF V5 is based on a different a priori database,  which, in contrast to the a priori database of GPROF V7, uses estimates from GPM DPR-Ku over land, this is expected.

**Reviewer comment 9**

Line 234: 'retrieval database' – which one is the retrieval database? I suppose you are referring to GPM CMB? This should be stated more clearly earlier in the section and be consistent throughout the manuscript.

**Author response:**

Again, thank you for pointing this out. We have revised the sentence to clearly state that the database we were referring to is the a prior/training database.

**Changes in manuscript:**

> **Changes starting in line 260:**
>
> Since the  a priori/training database is derived from  GPM CMB, GPM CMB is practically bias-free compared to the retrieval database.

**Reviewer comment 10**

Line 244: 'introduced rain gauge correction' – replace with 'introduced by the rain gauge correction'.

**Author response:**

We have fixed this in the updated version of the manuscript.

**Changes in manuscript:**

> **Changes starting in line 270:**
>
> Given that the CMB over land is largely a radar-derived product, it is possible that the bias relative to MRMS is introduced by the rain gauge correction applied to the validation data and caused by precipitation properties that may not be resolved by the radar observations.

**Reviewer comment 11**

Figure 2: I see a very interesting behavior in the low values trend lines. The GPM CMB vs a-priori dataset (which is GPM CMB 2019) have overestimation of the GPM CMB 2019 compared to GPM CMB 'other years', while all the others have the opposite behaviors. Also it looks like the comparison with MRMS 2021 and 2022 shows higher bias for low values. Also the trend for higher values is worth attention. It might be nice to reference this behavior in the section and in the bias description since it provides more information on the range of precipitation that has most issues.

**Author response:**

The opposing behavior of GPM-CMB and GPROF in the comparison against the a priori/training database, is likely due to the different resolution of the GPROF(-NN) and GPM CMB retrievals. GPM CMB having lower resolution than the precipitation in the a priori database causes and underestimation of light precipitation and overestimation of heavy precipitation.

It is true that there is a distinctive change in the conditional mean for light MRMS precipitation starting in the water year 2021. We suspect that this is connected to the change in the processing of the MRMS measurements that is also behind the change in the biases that occurrs at the same time.

We have revised the manuscript to discuss both of these points.

**Changes in manuscript:**

1. We have extended the discussion of the retrieval accuracy with respect to the a priori database:

   **Changes starting in line 238:**

   The spread in these distributions is due to the limitations of the retrieval method and the ill-posed character of the retrieval and they thus represent the best-case accuracy of the GPROF and GPROF-NN retrievals. Some spread is observed even between GPM CMB and the a priori database. This is due to the spatial smoothing applied to the precipitation in the a priori database, which causes precipitation measurements in the a priori/training database to have lower resolution than GPM CMB. The difference in the resolution between the GPROF(-NN) retrievals and GPM CMB with respect to the a priori database also explains the opposing behavior of the respective conditional means at light and heavy precipitation rates.

2. We have added the following sentence to the discussion of the inter-annual variability of the scatter plots:

   **Changes starting in line 248:**

   The distributions corresponding to different validation periods are very similar for all retrievals. An exception to this is a notable increase in the overestimation of light precipitation after the water year 2020 that affects all retrievals. As will be discussed below, this coincides with a change in the regional biases and is likely due to a change in the processing of the MRMS esimates (Anonymous Referee 2, 2023)

**Reviewer comment 12**

Line 260: 'conventional GPROF' – both V7 and V5?

**Author response:**

Yes, conventional GPROF is meant to refer to both GPROF V5 and V7. To avoid confusion, we have replaced 'conventional GPROF' with 'GPROF V5 and V7'.

**Changes in manuscript:**

> **Changes starting in line 290:**
>
> For the  GPROF V5 and V7 retrievals, correlations are between 0.4 and 0.6 over most of the CONUS.

**Reviewer comment 13**

Line 270: 'When compared to the database' – which database?

**Author response:**

Here we mean the a priori/training database. We have updated the manuscript to make this clear.

**Changes in manuscript:**

> **Changes starting in line 302:**
>
>  All retrievals exhibit weak biases of the order of a few percent compared to the a priori/training database.

**Reviewer comment 14 and 15**

Line 294: 'the fraction of confirmed raining pixels among those retrieved as raining' – would this be 'the fraction of confirmed raining pixels in the a priori database among those retrieved as raining by MRMS'?

Line 295: 'the fraction of confirmed raining pixels that are detected by retrieval' – would this be 'the fraction of confirmed raining pixels in the a priori database that are detected by retrieval'? I might have interpreted these last two sentences incorrectly which suggests the importance of clarifying which datasets you are talking about.

**Author response:**

We have revised the corresponding paragraph to clarify the definition of precision and recall as well as the role of the a priori database and MRMS validation data.

**Changes in manuscript:**

> **Changes starting in line 326:**
>
> The  , i.e., precision of the retrievals is the fraction of  truly raining pixels and the total number of pixels retrieved as raining , . The recall corresponds to the fraction of  actually raining pixels that are detected by  each retrieval. Here, *actually raining* is defined with respect to the assumed ground truth, which is the a priori/training database or MRMS validation data depending on which of the two sources the retrievals are compared against.

**Reviewer comment 16**

Line 294-295: you talk about raining pixels. So frozen precipitation is excluded also from GPROF? I mean, it makes sense, but you mentioned it is excluded from MRMS earlier in the manuscript and never mentioned what you are doing for GPROF or CMB. I think this is a big point since it eliminates a lot of winter observations that, together with the winter precipitation mentioned in the regional analysis, needs to be clarified earlier in the manuscript.

**Author response:**

Samples identified as frozen precipitation are excluded from most of our analyses from both the MRMS as well as the GPROF(-NN) results. For the identification of frozen precipitation we rely on MRMS because we deem it to be more reliable than corresponding classification derived from satellite retrievals.

We hope that the changes listed in response to the reviewer's second principal comment make this point clear.

**Changes in manuscript:**

See principal comment 2.

**Reviewer comment 17**

Line 308: 'For both the database and the MRMS' – do you mean the a priori database?

**Author response:**

Yes, we mean tha a priori database. We have rewritten this sentence to make this clear.

**Changes in manuscript:**

> **Changes starting in line 341:**
>
> The error metrics for the retrievals in the six regions are shown in Fig. 8. For both the a priori database and the MRMS  validation measurements, the regional biases are generally larger in magnitude than they are for the full CONUS.

**Reviewer comment 18**

Figure 6 caption: Panel (a) shows the detection skill for the database collocations – I would specify a priori database.

**Author response:**

We have adopted this suggestion in the revised manuscript.

**Changes in manuscript:**

We have updated the caption of Fig. 6:

> **Changes starting in line :**
>
> Precision-recall curves for the GPROF retrievals. Panel (a) shows the detection skill for the  collocations with the a priori/training database. Remaining panels show the results with respect to the MRMS validation measurements for the years 2019, 2020, 2021, and 2022.

**Reviewer comment 19**

Figure 12: for better comparison I would suggest to add a column with the GMI results in these plots.

**Author response:**

We have updated both Fig. 12 and 13 in the revised version of the manuscript, which now looks as shown in Fig. 1.2 and Fig. 1.3, respectively.

**Reviewer comment 20**

Line 464-465: I actually see more bias for GPROF V5 and V7 than from GPROF NN, am I missing something?

**Author response:**

Yes, the remaning biases are larger for GPROF V5 and V7 than for GPROF-NN. What we were trying to point out was that the *the reduction in bias* is more substantial for the

[Figure]

Figure 1.2: Accuracy metrics for a selection of other sensors of the GPM constellation evaluated against gauge corrected MRMS measurements of CONUS.

[Figure]

Figure 1.3: Accuracy metrics for a selection of other sensors of the GPM constellation evaluated against the K-POL radar in the tropical Pacific.

neural-network-based retrievals leading to overall lower biases after accounting for the database biases.

We have reformulated the offending section to avoid this kind of confusion.

**Changes in manuscript:**

**Changes starting in line 539:**

Subtracting the GPM CMB biases reduces the overall magnitude of the biases and their temporal and spatial variability. The  reduction in bias is more pronounced for the GPROF-NN retrievals indicating that their higher accuracy puts increased weight on the reference data.

**Bibliography**

Anonymous Referee 2 (2023). Comment on egusphere-2023-1310. `https://doi.org/10.5194/egusphere-2023-1310-RC2`. Accessed: 2023-10-08.

Huffman, G. J., Bolvin, D. T., Braithwaite, D., Hsu, K.-L., Joyce, R. J., Kidd, C., Nelkin, E. J., Sorooshian, S., Stocker, E. F., Tan, J., Wolff, D. B., and Xie, P. (2020). *Integrated Multi-satellite Retrievals for the Global Precipitation Measurement (GPM) Mission (IMERG)*, pages 343–353. Springer International Publishing, Cham.

---

## Author Comment (AC2)

**GPROF V7 and beyond: Assessment of current and potential future versions of the GPROF passive microwave precipitation retrievals against ground radar measurements over the continental US and the Pacific Ocean**

**Response to reviewers**

**1 Response to reviewer 1**

We thank the reviewer for investing their time to read our manuscript and provide constructive feedback to improve it. The principal changes that we have implemented based on comments from both reviewers are the following:

1. We repeated our analysis excluding all precipitation classified as frozen by MRMS, i.e., excluding not only snow but also hail. Significant effects of excluding hail were only observed in the SW region and we have updated the manuscript accordingly.

2. We have added a short section that analyzes the impact of excluding both frozen precipitation as well as precipitation of snow-covered and mountain surfaces and the behavior of the different retrievals over those surfaces.

3. We have revised all references to the a priori database and the MRMS validation data to make the distinction of the two clearer.

Finally, we have also updated that precision-recall curves to show the precision over the recall instead of the recall over the precision, which is the more common way to present these curves. During this, we also realized that the PR curves previously included samples over ocean, which we have now excluded. This lead to a minor change in the relative skill of the GPROF-NN HR retrieval.

In what follows, line and figure numbers are given with respect to the revised manuscript.

**1.1 Specific comments**

**Reviewer comment 1**

Abstract, Line 12-13: What does "retrieval reproduces the principal precipitation characteristics of each region" mean? Can you please elaborate?

**Author response:**

What we were referring to here were the different regional seasonal and diurnal cycles of precipitation, which were reliably reproduced by the GPROF retrievals. We have reformulated the sentence in the revised manuscript to make this clear.

**Changes in manuscript:**

**Changes starting in line 13:**

Although biases of up to 25 % are observed over sub-regions of the CONUS and the tropical Pacific, the retrieval reliably reproduces each region's diurnal and seasonal precipitation characteristics.

**Reviewer comment 2**

Abstract, Line16-18: I appreciate that authors are providing this significant finding here at the abstract, however, can you please be more specific about the time resolution of this comparison? Meaning at what time resolution GPROF NN 1D is improving mean absolute error, corr etc.?

**Author response:**

All retrieval errors are computed with respect to instantaneous precipitation estimates at 5 km resolution. We have added this information to the sentence in question in the revised version of the manuscript.

**Changes in manuscript:**

**Changes starting in line 18:**

GPROF-NN 1D, the most basic neural network implementation of GPROF, improves the mean-squared error, mean absolute error, correlation and symmetric mean absolute percentage error of instantaneous precipitation estimates by about twenty percent for GPROF GMI while the effective resolution is improved to 31 km over land and 15 km over oceans.

**Reviewer comment 3**

Line 58-60: This is a very confusing sentence. Can you please reword it?

**Author response:**

We agree with the reviewer that the sentence is badly worded. We have reformulated it in the revised version of the manuscript.

**Changes in manuscript**

**Changes starting in line 58:**

This study compares  GPROF retrievals to independent  validation data derived from ground-based precipitation radars. In

this case, differences between a priori database and  the validation data constitute a second source of errors that will increase the total retrieval error. These two sources of error are fundamentally different and reducing their impact requires different approaches. Therefore, quantifying the extent to which these sources contribute to the total retrieval error is essential to guide future efforts to improve GPM PMW retrievals.

**Reviewer comment 4**

Line 64: Can you please reword this question, something along the lines: "to what extent a priori database errors contribute to GPROF overall retrieval errors?"

**Author response:**

We have reworded the question in the revised version of the manuscript.

**Changes in manuscript**

**Changes starting in line 66:**

 What is the contribution of errors in the a priori database to the GPROF retrieval error?

**Reviewer comment 5**

Line 65: I think it would be better to remove "even" from this question"... GPM PMW observations even when compared to ..."

**Author response:**

We have removed even in the revised version of the manuscript.

**Changes in manuscript**

**Changes starting in line 67:**

Can the GPROF-NN retrievals improve precipitation estimates of the GPM PMW observations  when compared to independent measurements?

**Reviewer comment 6**

Line 90: Authors mention that rain gauge corrected MRMS data are used. Can authors please be more specific which database they have used? Because the way it has been presented is slightly confusing. Gauge corrected MRMS precipitation magnitudes are accumulations. However, radar only MRMS data provides precipitation rates at 2 min temporal intervals. Did the authors conduct their own gauge correction to the radar only MRMS product?

**Author response:**

The MRMS estimates that are used in the study are instantaneous, gauge-corrected radar QPEs. This is a special product that is produced specifically for GPM ground validation. It uses gauge correction factors derived from hourly gauge-corrected and radar-only accumulations to correct instantaneous radar QPE's.

**Changes in manuscript:**

> **Changes starting in line 97:**
>
>  The principal source of validation measurements for this study are instantaneous, gauge corrected precipitation estimates from the NOAA Multi-Radar Multi-Sensor System (MRMS). These estimates are produced specifically for GPM ground validation and are gauge-corrected to match monthly accumulations  following the approach described (Kirstetter et al., 2012). These estimates are not part of the operational MRMS processing suite but can be obtained from the GPM ground validation data archive (Wolff, 2023). The processing of the ground-validation data includes a basic filtering that removes measurements with excessive gauge-correction factors (Kirstetter et al., 2012). The data is provided on an approximately $0.01\,° \times 0.01\,°$ grid covering  the CONUS. For the comparison against the satellite retrievals, the MRMS data is smoothed using a Gaussian average filter with a full-width at half-maximum (FWHM) of 5 km. Following this, the mapping to the collocation grid is performed using nearest-neighbor interpolation.

**Reviewer comment 7**

Line 211-212: Can authors please clearly indicate whether the mountain surfaces are excluded or included with a correction.

**Author response:**

We have rewritten this section to clearly state that these pixels are excluded because of the correction applied to them.

**Changes in manuscript**

> **Changes starting in line 226:**
>
>  Retrievals over snow-covered and mountain surfaces are excluded from the validation  due to the uncertainties in both the satellite estimates as well as the validation data. In addition to this, the GPROF a priori database for snow-covered surfaces is derived from collocations with MRMS, while precipitation over mountains is obtained by scaling the GPM CMB precipitation to account for the orographic enhancement of precipitation. These two modifications aim to counteract known weaknesses  of the GPM CMB  retrievals, but would skew the comparison between GPM, GPM CMB, and MRMS.
>
> Similarly, precipitation that is identified as  frozen by MRMS is excluded from the validation. The retrieval of frozen precipitation from both PMW and radar is particularly challenging due to its uncertain radiometric properties. Because of these increased uncertainties and the small contribution of frozen precipitation to the total precipitation in the validation data, the retrieval accuracy for frozen precipitation should be assessed in a dedicated study.

**Reviewer comment 8**

Line 228: Can authors please explain how they calculate the bias or what is the definition of the bias? And at what temporal resolution (I am assuming this is annual but it would be nice to indicate).

**Author response:**

We have added the requested information in the revised version of the manuscript.

**Changes in manuscript**

> **Changes starting in line 254:**
>
> Maps of annual mean retrieval biases, calculated as the annual average of the difference between retrieved precipitation and the precipitation in the a priori database or the MRMS validation data, are displayed in Fig. 3.

**Reviewer comment 9**

Line 239-242: To add to the explanation here (this is through a personal communication with a MRMS team member): "It is not documented on Iowa website, however, in Oct 2020, the gauge correction methodology and associated products are changed." This corresponds exactly to the 2021 water year that authors are using in this study.

**Author response:**

We would like to thank the reviewer for this useful information. We have included it in the revised version of the manuscript.

**Changes in manuscript**

> **Changes starting in line 274:**
>
>  As pointed out by one of the anonymous reviewers, it is likely that this is due a change in the  gauge correction methodology that occurred around October 2020 (Anonymous Referee 2, 2023) .

**1.1.1 Reviewer comment 10**

Line 244: "... it is possible that the bias relative to MRMS is introduced by rain gauge correction" please include "by" in this sentence to make it clear.

**Author response:**

We have fixed this in the updated version of the manuscript.

**Changes in manuscript:**

> **Changes starting in line 270:**
>
> Given that the CMB over land is largely a radar-derived product, it is possible that the bias relative to MRMS is introduced by the rain gauge correction applied to the validation data and caused by precipitation properties that may not be resolved by the radar observations.

**Reviewer comment 10**

Line 266-268: Can authors please describe why they decide to use mean error, mean-squared error and mean absolute error all together? What do they explain differently and why did authors needed all of them together? Moreover, can authors please describe symmetric mean absolute percentage error in more detail i.e., what does this score mean, what are the max and min values etc.

**Author response:**

We have decided to include multiple error metrics in our analysis because our ultimate aim is to improve precipitation estimates and not just tune them to optimize a single error metric. Both MSE and MAE are fairly common error metrics and providing them

can provide a reference for other retrievals. However, MSE and, to a lesser extent, MAE are dominated by heavy precipitation.

A relative error such as the mean absolute percentage error (MAPE) is more sensitive retrieval errors for light precipitation estimates (c.f. Fig. 1.2). However, the issue with the MAPE is that it penalizes overestimation heavier than underestimation. The symmetric mean absolute percentage error corrects this shortcoming by modifying the MAPE to be symmetric in its arguments.

We added an appendix to the revised manuscript that motivates our choice of metrics, states their formulas, and illustrates the characteristics of MSE, MAE, MAPE, and SMAPE. The appendix contains the figure shown in Fig. 1.1, which displays the different behavior of the error functions underlying MAE, MSE, MAPE and SMAPE. It also shows the asymmetry of the MAPE, which motivated our choice of the SMAPE over MAPE.

Finally, the appendix also contains the figure shown in Fig. 1.2, which shows the relative contribution of different precipitation intensities to the final value of the metric. This figure clearly shows the complementary behavior of MSE, MAE, and SMAPE in terms of sensitivity to different precipitation intensities.

**Changes in manuscript:**

1. We have rewritten the paragraph introducing the metrics:

   **Changes starting in line 297:**

[revised manuscript text omitted]

---

## Author Response (AR2)

**GPROF V7 and beyond: Assessment of current and potential future versions of the GPROF passive microwave precipitation retrievals against ground radar measurements over the continental US and the Pacific Ocean**

Response to reviewer comments

This document contains the responses to each reviewer's comments. For each comment, the author's response and, if applicable, the corresponding changes in the manuscript are listed. Line numbers of changes are given with respect to the revised manuscript.

We thank both reviewers for investing their time to read our revised manuscript and making us aware of outstanding typographical errors.

**1 Comments from reviewer 1**

In what follows, line and figure numbers are given with respect to the revised manuscript.

**1.1 Typographical errors**

**Reviewer comment 1**

l. 75 remove this in "The principal motivation this is that PMW..."

**Author response:**

Instead of removing 'this' we suggest to insert 'for' to make clear what the motivation refers to.

**Changes in manuscript:**

> **Changes starting in line 71:**
>
> The principal motivation for this is that PMW precipitation estimates of frozen precipitation and over snow-covered and mountain surfaces are particularly uncertain, and, since they constitute only a minor part of the total validation data, corresponding larger retrieval errors may not be reflected in the overall validation statistics.

**Reviewer comment 2**

l.105 I would replace "described (Kirstetter et al 2012)" with "described in Kirstetter et al 2012".

**Author response:**

We have corrected this in the revised version of the manuscript.

**Changes in manuscript:**

> **Changes starting in line 99:**
>
> These estimates are produced specifically for GPM ground validation and are gauge-corrected to match monthly accumulations following the approach described (Kirstetter et al., 2012) by Kirstetter et al. (2012).

**Reviewer comment 3**

l. 190-191 redundant: "The distance between the centers of neighboring pixels of the GMI swath is approximately 13.5 km in along-track direction and 5 km in across-track directions at the center of the swath". I suggest to replace with: "The distance between the centers of neighboring pixels of the GMI swath is approximately 13.5 km in along-track direction and 5 km in across-track directions".

**Author response:**

We thank the reviwer for this suggestion. We will also replace 5 km with 6 km because that is closer to the actual across-track distance between pixel centers.

**Changes in manuscript:**

> **Changes starting in line 184:**
>
> The distance between the centers of neighboring pixels of the GMI swath is approximately 13.5 km in along-track direction and  6 km in across-track direction.

**Reviewer comment 4**

L.286 0.5deg seems a bit more than 5 km. Should it be 0.05? But maybe 5 km in this case is just fine since you introduced the collocated resolution as 5km.

**Author response:**

We want to thank the reviewer for pointing out this inconsistency. In fact, the sentence in question referred to the resolution of the bias maps, which is 5 ° and not 0.5 °. We have reformulated the sentence to make this clear and inserted the correct resolution.

**Changes in manuscript:**

> **Changes starting in line 184:**
>
>  Aggregated to the 5° resolution used for the analysis presented in Fig. 3, the biases of GPROF V7 and the GPROF-NN retrievals are well correlated with those of GPM CMB and exhibit similar variability throughout the assessed years.

**Reviewer comment 5**

l.379 correct prevision with previous

**Author response:**

We have corrected this in the revised version of the manuscript.

**Changes in manuscript:**

**Changes starting in line 352:**

The increase in ocean precipitation for GPROF V7 and the GPROF-NN retrievals can be explained by the inclusion of light precipitation from MIRS where GPM CMB doesn't detect rain and an increase in ocean precipitation in GPM CMB V7 compared to the  previous version of GPM CMB, upon which the GPROF V5 apriori database was based on over ocean surfaces.

**2 Comments from reviewer 2**

Reviewer 2 did not request any changes to the manuscript.

**Bibliography**

Kirstetter, P.-E., Hong, Y., Gourley, J. J., Chen, S., Flamig, Z., Zhang, J., Schwaller, M., Petersen, W., and Amitai, E.: Toward a Framework for Systematic Error Modeling of Spaceborne Precipitation Radar with NOAA/NSSL Ground Radar-Based National Mosaic QPE, Journal of Hydrometeorology, 13, 1285 – 1300, https://doi.org/10.1175/JHM-D-11-0139.1, 2012.